

# MIPAS observations of longitudinal oscillations in the mesosphere and the lower thermosphere: Part 1. Climatology of odd-parity daily frequency modes

Maya García-Comas[1], Francisco González-Galindo[1], Bernd Funke[1], Angela Gardini[1], Aythami Jurado-Navarro[1], Manuel López-Puertas[1], and William E. Ward[2]

[1]Instituto de Astrofísica de Andalucía-CSIC, Granada, Spain
[2]Department of Physics, University of New Brunswick, Fredericton, New Brunswick, Canada

*Correspondence to:* Maya García-Comas
Instituto de Astrofísica de Andalucía-CSIC
Glorieta de la Astronomía s/n
18008 Granada, Spain (maya@iaa.es)

**Abstract.** MIPAS global sun-synchronous observations are almost locked in local time. Subtraction of the descending and ascending node measurements at each longitude only contain the longitudinal oscillations with odd daily frequencies $n_{odd}$ from a solar perspective at 10 A.M. Contributions of the background atmosphere, persistent (on a daily basis) longitudinal oscillations and tidal modes with even daily frequencies vanish. We have determined MIPAS temperature longitudinal oscillations with

$n_{odd}$ and wavenumber $k=0-4$ from 20 to 150 km from April 2007 to March 2012. To our knowledge, this is the first time temperature zonal oscillations are derived in this altitude range globally from a single instrument. The major findings are the detection of: 1) migrating tides at Northern and Southern high latitudes; 2) significant $k = 1$ activity at extra-tropical and high-latitudes, particularly in the SH; 3) $k = 3$ and $k = 4$ eastward propagating waves that penetrate in the lower thermosphere with a significantly larger vertical wavelength than in the mesosphere; 4) a quasi-biennial oscillation of the migrating tide mainly

originated in the stratosphere and propagated to the MLT. MIPAS global measurements of longitudinal oscillations are useful for testing tide modeling in the MLT region and as a lower boundary of models extending higher up in the atmosphere.

## 1 Introduction

One of the most prominent and persistent dynamical features in the mesosphere and the lower thermosphere (MLT) are the atmospheric solar tides (referred as simply *tides* hereafter). They are oscillations with periods that are subharmonics of a solar

day. Particular cases among them are the migrating tides are westward propagating oscillations with a phase speed equal to the Earth's angular velocity, $\Omega$. They apparently travel with the Sun, and depend on the solar local time (LST) but not on longitude. Migrating tides are excited by a longitude independent source, such as the absorption of solar IR radiation by water vapor in the troposphere, the solar UV radiation absorption by ozone in the stratosphere, and the local solar UV and EUV radiation absorption by oxygen molecules and atoms, respectively, and maybe also chemical heating (Smith et al., 2003), in

the thermosphere.





The non-migrating tides also have periods which are subharmonics of a solar day but may be either westward propagating, or stationary and their phase speed is different from $\Omega$. They are excited by longitudinally varying properties, like tropospheric latent heat release from evaporation, Sun gravitational pull or heating rates, and by nonlinear wave-wave interactions. An interaction between zonal wavenumber $s$ asymmetries in surface or atmospheric properties and the absorption of the $n^{th}$ harmonic of the diurnally varying solar radiation generates a sum and a difference tide with frequency $n\Omega$ and zonal wavenumbers $n \pm s$. For example, the zonally asymmetric latent heat release in the tropical troposphere caused by the wavenumber-4 land-sea distribution modulates the migrating diurnal DW1 component to excite the DE3 and DW5 tidal pair (Hagan and Forbes, 2002; Zhang et al., 2006). As an example of wave-wave interactions, involving tides and planetary waves, the interaction of the stationary planetary wave-1 sPW1 with DW1 leads to the formation of D0 and DW2 tidal modes (Hagan and Roble, 2001). Note that the widely used 3-character notation for tides is also used here. The first letter corresponds to the daily frequency (D corresponds to diurnal; S corresponds to semidiurnal; T corresponds to terdiurnal). The second letter indicates the direction of propagation (W for westward and E for eastward). The number is the absolute value of the tide zonal wavenumber.

Tides also interact with other dynamical processes. Tidal inter-annual variability is thought to be correlated with the El Nino-Southern Oscillation (ENSO) (Gurubaran et al., 2005) and the wind Quasi-Biennial Oscillation (QBO) (McLandress, 2002; Mayr and Mengel, 2005); planetary and gravity wave (GW) activity may affect tidal activity (Fritts and Vincent, 1987; Teitelbaum and Vial, 1991; Pedatella and Liu, 2012; Ribstein et al., 2015); GW and tidal non-linear interaction significantly affects winds (Liu et al., 2014). Nevertheless, there is no complete understanding of how these processes take place. For example, it is not clear if the mesospheric QBO signature originates at stratospheric levels, locally in the mesosphere or both (Oberheide et al., 2009).

Tides generally propagate upwards. Despite the location of its source, the amplitude of upward propagating tides grows with altitude due to the conservation of energy in a density decreasing with altitude environment. The tides originated at the troposphere that propagate vertically connect the lower, the middle and the upper atmospheres. They also produce a second order impact on atmospheric vertical coupling as they modulate the upward propagation of other waves, like gravity waves (Eckermann and Marks, 1996; Senf and Achatz, 2011). The connection also extends in the latitudinal direction because tides become a global feature under its standing wave latitudinal structure.

The extent to which tides propagate from the lower atmosphere to the thermosphere or to which changes in lower altitude regions are transmitted by tides to the upper atmosphere or to other latitudes is not completely known. The 100-150 km range is a region of particular interest. Vertically-propagating tides maximize at those altitudes, where molecular dissipation dominates (Forbes and Garrett, 1979). They impact the spatial and temporal variability through electrodynamical effects, that are transmitted even higher (Kil et al., 2007; Hagan et al., 2007; Jin et al., 2008). The temperature tidal spectrum at these altitudes is not well known. Few global observations of the neutral atmosphere are available there. TIMED measurements covered altitudes below 115 km (see e.g., Xu et al., 2009; Pancheva and Mukhtarov, 2011; Yue et al., 2013; Truskowski et al., 2014) and CHAMP above 400 km (Forbes et al., 2008; Oberheide et al., 2009; Bruinsma and Forbes, 2010; Forbes et al., 2014, see e.g., ). Besides the knowledge of the local behavior, observations at 100-150 km are needed to discern the origin of longitudinal





oscillations higher up in the atmosphere. Characterizing the non-migrating tides is relevant even only in the lowest altitudes of this range, as a lower boundary in numerical models.

The analysis of tides from atmospheric measurements is generally complicated. Ideally, the latitudinal, longitudinal and local time coverage should be complete to isolate the tidal components contributing to the observed signal. The absence of
longitudinal coverage of ground-based instrumentation results in strong tidal aliasing. Analogously, sun-synchronous satellite instrumentation (with two fixed local times of observations 12 hours apart) does not provide a good local time coverage but, in contrast to ground-based instruments, it allows for tidal observations on a global scale. Besides, the combination of the measurements taken at these two fixed times cancels out particular tidal frequencies and reduces aliasing, as we do in this work. Non-sun-synchronous instruments certainly allow for the separation of individual tidal components (Zhang et al., 2006;
Gan et al., 2014; Truskowski et al., 2014). However, their slow preceeding period requires grouping measurements covering periods larger than months in order to achieve a complete local time coverage and, thus, the temporal resolution is degraded. Isolation of the tidal components with a finer time resolution can be achieved only if two tidal components with daily frequency of different parity contribute to an observed longitudinal wavenumber in the local time frame (Li et al., 2015). Thus, as we will show here, the advantage of sun-synchronous instruments is their ability to better resolve longitudinal oscillations globally at
high time resolution, particularly, where more than two daily frequencies contribute.

The Michelson Interferometer for Passive Atmospheric Sounding (MIPAS) (Fischer et al., 2008) measured the Earth temperature globally from pole-to-pole covering altitudes from 20 to 150 km onboard a sun-synchronous satellite, Envisat. Observations of temperature longitudinal oscillations with MIPAS wide spatial (both horizontal and vertical) and temporal coverage are ideal for constraining tide vertical and latitudinal extent and global variations in seasonal and inter-annual timescales.
Therefore, they provide a wealth of information on the tidal excitation mechanisms, the processes inducing tidal variability and the lower and upper atmosphere coupling through tides. Additionally, the characterization of the tides in an atmospheric measurement dataset is a prerequisite to estimate trends (Beig et al., 2003).

This work describes the MLT temperature longitudinal oscillations measured by MIPAS during five full years (March 2007-March 2012). Inspired by its sun-synchronous observational strategy, we extract the longitudinal wavenumbers as would be
viewed by an observer sitting on the Sun with the ability to isolate daily frequency parities. We divided the work in two parts: this paper covers the observed odd daily frequency temperature components and a companion paper focuses on the even daily frequency oscillations (García-Comas et al., 2015).

This paper is organized as follows. Section 2 describes the MIPAS observations used in this work. Section 3 presents the method used to extract longitudinal oscillations from a longitudinal series of observations. The monthly climatologies of the
observed longitudinal wavenumbers are presented in Sect. 4, where we discuss their latitude, altitude and seasonal behavior in the context of other measurements. Section 5 includes a description of the inter-annual variability of the main oscillations. We close up with a summary of the main findings.





## 2 The instrument, data set and error sources

MIPAS measurements are distributed over the full globe and were taken during the descending and the ascending nodes at
approximately two fixed local times, 10 A.M. and 10 P.M., respectively (in general, local times differ from the average values
less than half an hour at latitudes equatorward of $75°$ and three quarters of an hour equatorward of $85°$). The MIPAS Middle
Atmosphere (MA) and the Upper Atmosphere (UA) modes of observation resulted in an altitude coverage of 18–102 km
and 40–170 km, respectively (Oelhaf, 2008).While operating with optimized-resolution ($0.0625\,cm^{-1}$; unapodized), full day
observations using these modes were performed regularly (approximately one day in each mode every 10 days) from April
2007 to March 2012, thus covering five complete years.

MIPAS spectra covered from 4.3 to $15.6\,\mu m$. Temperature and line of sight information (LOS) are derived in the MLT re-
gion from the $15\,\mu m$ $CO_2$ emission using the IMK/IAA scientific MIPAS level 2 processor described in von Clarmann et al.
(2009), which accounts for non-LTE effects using the GRANADA model (Funke et al., 2012). We use in this work data versions
V5R_TLOS_521 and V5R_TLOS_621 of MA and UA, respectively, retrievals (García-Comas et al., 2014), which provide tem-
perature profiles with 3 to 10 km vertical resolution from the stratosphere up to around 110 km ($T_{MLT}$) (García-Comas et al.,
2012).

Thermospheric temperatures ($T_{THER}$) from 115 km up to 150 km are derived from MIPAS $5.3\,\mu m$ NO emission measure-
ments in the UA mode (Bermejo-Pantaleón et al., 2011) with a 5-10 km vertical resolution. At the time of writing this paper, a
single data version covering the 2007-2012 period is not yet available. We therefore use two retrieval versions, V4O_TT_611
(2007-2009) and V5R_TT_621 (2010-2012) of $T_{THER}$. These two data versions differ mainly in the spectra calibration ver-
sion supplied by ESA. Bermejo-Pantaleón et al. (2011) found that their derived nighttime temperatures (V4O_TT_611) might
be affected by artifacts. The new data version V5R_TT_621 corrects this problem by using NOEM day and night concentra-
tions (Marsh et al., 2004) as *a priori*. Tests showed that the new temperature version leads to similar structures of the day/night
zonal mean differences but smaller in absolute value by 10-15K below 140 km and 15-25K above 140 km. It is important noting
that, even if the day/night temperature differences retrieved from these versions do not differ, their longitudinal anomalies are
similar (amplitude differences smaller than 5K). Thus, combination of these two versions is reasonable, in particular, for the
study of non-migrating components.

Despite MIPAS temperature systematic errors (1-3K below 85 km and 3-10K above 85 km) for $T_{MLT}$, and 10-20K for
$T_{THER}$), we analyze here differences of longitudinal perturbations about an average value (see Sect. 3). Thus, biases often
partially cancel out so that their effect on the derived oscillations characteristics are small. Additionally, we work with monthly
averages, which reduce the random error by at least a factor of $1/\sqrt{3}$. One of the most significant errors we expect is that
produced by the limited vertical resolution. At the peaks of a vertical wave, the wave amplitude error due to that smoothing
$A_{smoo}$ can be estimated with:

$$\Delta A_{smoo} = A\Big(1 - \frac{\lambda_z}{\pi \Delta z} \sin \frac{\pi \Delta z}{\lambda_z}\Big), \tag{1}$$





where $A$ is the wave amplitude, $\lambda_z$ is the wave vertical wavelenght, and $\Delta z$ is the vertical resolution. As an example, MIPAS $T_{MLT}$ smoothing error for a $10\,\mathrm{km}$ vertical wavelength wave is 15% whereas for a $30\,\mathrm{km}$ vertical wavelength wave the error is 2%.

The temperature retrieval algorithm uses *a priori* information of the temperature at each MIPAS geolocation taken from ECMWF at pressures larger than $0.1\,\mathrm{hPa}$ and merged with NRLMSISE-00 model results at lower pressures.NRLMSISE-00

includes only effects of low order tides (Picone et al., 2002). In order to detect any potential impact of *a priori* zonal oscillations on the retrieved temperatures, we examined NRLMSISE-00 temperature variations with longitude in the fixed local time frame of MIPAS. A migrating component most likely corresponding to DW1 is present in the model, with the largest perturbations in the equinoxes and over the equator and alternating minima and maxima at $55\,\mathrm{km}$, $70\,\mathrm{km}$, $95\,\mathrm{km}$ and $110\,\mathrm{km}$. Out of phase weaker oscillations appear only at $60°$ North and South at $70\text{-}75\,\mathrm{km}$. Another maximum appears above $145\,\mathrm{km}$, displaced

$20\text{-}30°$ off the equator in the summer, that is, following the sub-solar point. We only detect wavenumber 1 and 2 non-migrating zonal oscillations with amplitudes generally smaller than $1K$. Some exceptions occur in the winter high latitudes, probably corresponding to planetary wave activity (wavenumber-1 structures reaching $10K$ at $40\,\mathrm{km}$, $5K$ at $50\,\mathrm{km}$ and $2K$ at $75\,\mathrm{km}$).

## 3 Methodology

Considering only the effect of tides, an atmospheric variable $X$ (as temperature or concentration of certain species) at altitude

$z$, latitude $\phi$, longitude $\lambda$ and Universal Time (UT) $t'$ can be expressed as a perturbation of the background zonal mean value of the variable, $\overline{X}$, by the sum of all individual tidal components $X'_{n,s}$ with zonal wavenumber $s$ and wave frequency $n$ at that position and time:

$$X(\lambda, t') = \overline{X} + \sum_{n,s} X'_{n,s}(\lambda, t'), \tag{2}$$

where we have suppressed the dependence of $X$, $\overline{X}$ and $X'_{n,s}$ on $z$ and $\phi$ for clarity. Each tidal component can be expressed as

a sinusodial longitudinal oscillation of the form:

$$X'_{n,s}(\lambda, t') = A_{n,s} \cos\left(n\Omega t' + s\lambda + \Phi_{n,s}\right), \tag{3}$$

where $\Omega = 2\pi/24$ is the Earth's angular velocity, $A_{n,s}$ is the amplitude of the component and $\Phi_{n,s}$ is the oscillation phase. We note that $A_{n,s}$ and $\Phi_{n,s}$ are also functions of $z$ and $\phi$. The zonal wavenumber is defined such that $s < 0$ refers to eastward propagating tides, $s > 0$ for westward propagating tides and $s = 0$ for stationary tides. The wave frequency $n$ is a positive

integer such that $n = 1$ for a diurnal component, $n = 2$ for a semi-diurnal component, and successively.

UT and local solar time, $t$, are related through $t' = t - \frac{24}{2\pi}\lambda$. Thus, Eq. 3 can also be written as:

$$X'_{n,s}(\lambda, t) = A_{n,s} \cos\left(n\Omega(t - \varphi_{n,s}) - (n - s)\lambda\right). \tag{4}$$

where $\varphi_{n,s}$ is now the phase defined as the local solar time for maximum amplitude at $\lambda{=}0$. For the particular case of a solar migrating tide, $s = n$ and Eq. 4 reduces to:

$$X'_{n,n}(t) = A_{n,n} \cos\left(n\Omega(t - \varphi_{n,n})\right), \tag{5}$$





which does not depend on $\lambda$ but only on local time $t$.

For a sun-synchronous instrument, the local solar time of the ascending and descending segments, $t_a$ and $t_d$, are fixed and $t_a = t_d + 12$. Therefore, using Eq. 4, a tidal perturbation at longitude $\lambda$ and local time $t_a$ is related to that at $t_d$ through:

$$X'_{n,s}(\lambda, t_a) = A_{n,s}\cos\left(n\Omega(t_a - \varphi_{n,s}) - (n-s)\lambda\right) =$$
$$= A_{n,s}\cos\left(n\Omega(t_d - \varphi_{n,s}) + n\pi - (n-s)\lambda\right) =$$
$$= (-1)^n X'_{n,s}(\lambda, t_d). \tag{6}$$

Subtracting the variable measured at each longitude $\lambda$ at the descending node local time $X(\lambda, t_d) = \overline{X} + \sum_{n,s} X'_{n,s}(\lambda, t_d)$ and that at the ascending node local time $X(\lambda, t_a) = \overline{X} + \sum_{n,s} X'_{n,s}(\lambda, t_a)$, dividing by two and using Eq. 6:

$$\Delta X/2 = \frac{X(\lambda, t_d) - X(\lambda, t_a)}{2} = \sum_{n-odd,s} X'_{n,s}(\lambda, t_d), \tag{7}$$

where addends with odd $n$ remain and those with even $n$ cancel out. We note that not only tidal components with even $n$ are removed but also the daily persistent longitudinal oscillations ($n = 0$), like planetary waves.

For each $z$ and $\phi$, $\Delta X/2$ in Eq. 7 is a function of longitude only that can be Fourier-decomposed in $\sum_k C_k \cos(k\lambda - \theta_k)$, $k$ being a positive integer. The change of variable $k = |n - s|$, with $n$ being an odd integer, in Eq. 4 and substituting in Eq. 7 leads to:

$$\sum_k C_k \cos(k\lambda - \theta_k) =$$
$$= \sum_k \sum_{n-odd} A_{n,n\pm k}\cos\left(n\Omega(t_d - \varphi_{n,n\pm k}) \pm k\lambda\right). \tag{8}$$

Since the sum of sinusoidal oscillations of the same frequency (in absolute value) provides a sinusoidal oscillation of that

frequency, we can make a correspondence between each $k$ term at the right-hand side of Eq. 8 with the corresponding one at the left-hand side. In other words, for a fixed LST sampling, all modes with $n$ being odd and $k = |n - s|$ appear to have a $2\pi/k$ zonal wavelength on the data and so does their combination.

The solution amplitudes $C_k(z, \phi)$ and phases $\theta_k(z, \phi)$ for the observed zonal wavenumber $k$ in $\Delta T/2$ embed the contributions from tidal oscillations $X'_{n,s}$ with $n$ being odd and such that $|n - s| = k$. We hereafter denote such combined oscillation

with wavenumber $k$ with $n-odd$ as $|n_{odd} - s| = k$. Table 1 summarizes the major tidal components contributing to the apparent zonal wavenumbers. A method similar to this one was successfully applied to CRISTA data initially by Oberheide et al. (2002).

Analogously to the subtraction, the sum of the variable pairs measured at the descending and at the ascending node LSTs at longitude $\lambda$ allows for the determination of the $n - even$ modes, which are the subject of the companion work of García-Comas et al. (2015).

Even if the tidal modes with the same $n$ parity and the same $|n - s|$ are aliased, a close look at the derived combined amplitudes $C_k$ and phases $\theta_k$ and their dependence with altitude could eventually help to discern the dominance of eastward propagating, westward propagating or standing tides.





There is an additional limitation for the derived information for $k = 0$ (migrating tides with $n - odd$) even if only one mode dominates. Half the amplitude $C_0$ derived from the Fourier fit corresponds to the term in Eq. 5 for $n - odd$. Thus, the amplitude $A_{n,n}$ of the tidal component can only be derived at the altitudes where the phase $\varphi_{n,n}$ happens at $t_d$ or $t_a$, which show up as maxima in the derived $C_0$ vertical profile. Another piece of information comes from the vertical nodes of that profile, corresponding to altitudes for which the phase $\varphi_{n,n}$ happens at $t_d \pm 6$.

Li et al. (2015) present a similar method but applied to any two fixed local times for the descending and the ascending nodes. However, the isolation of individual tidal components is then only possible when only two daily frequencies $n$ contribute, in their case $n = 1$ (diurnal) and $n = 2$ (semidiurnal). The advantage of the two fixed-local-time measurements being 12-hours-apart, as we use here, is that the separation of the $n - odd$ components from the $n - even$ ones is always possible, even if more than two daily frequencies contribute. That also includes $n=0$ (e.g., sPWs) or $n=3$ (terdiurnal), which are often a significant contribution.

## 4 Five-year average zonal oscillations

We have constructed longitude-altitude maps of monthly $T_{MLT}$ and $T_{THER}$ temperatures averaged from April 2007 to March 2012 for the descending and the ascending legs. In order to achieve a good longitudinal coverage with a small loss of horizontal resolution, we have used longitude and latitude running-means at each altitude in a $5° \times 5°$ grid. At each grid point, we have averaged MIPAS monthly measurements taken within $25°$ in longitude and $10°$ in latitude bins. We have then subtracted and divided by 2 the descending and the ascending node measurements at each grid point, $\Delta T/2$ hereafter.

Figure 1 left panels show typical $\Delta T/2$ fields. They are August and October monthly means averaged over the five years and constructed from $T_{MLT}$ and $T_{THER}$ measurements over the equator. A background with alternating positive and negative horizontal stripes is evident. A longitude-independent signature at each altitude must be responsible for that pattern. A longitude-independent feature in $\Delta T/2$ corresponds to a migrating component. The pattern is stronger during October (equinox) than August (solstice). Vertically alternating $\Delta T/2$ maxima and minima correspond to altitudes where the tidal phase is at 10 A.M. (45, 75, 95 and 110 km) and 10 P.M. (60, 85 105 and 120 km), respectively. A different regime takes place above 125 km, where a background positive $\Delta T/2$ remains nearly constant with altitude. That implies a nearly constant-with-altitude LST phase, that is, the oscillation does not propagate vertically. This is typical of an *in situ* generated oscillation.

The right panels in Fig. 1 are $\Delta T/2$ longitudinal anomalies (zonal mean subtracted at each altitude) averaged for August and October. They contain only $n - odd$ non-migrating tidal modes ($|n_{odd} - s|$ larger than 0). In August, an eastward propagating oscillation with wavenumber 4 is noticeable from the mid-mesosphere to the thermosphere with increasing amplitude up to 130 km. A subtle underlying wave-1 perturbation propagates to the west in the upper mesosphere but suffers efficient dissipation in the lowermost part of the thermosphere.

The October $\Delta T/2$ anomaly field in Fig. 1 exhibits a vertically (18 km vertical wavelength) and westward propagating wavenumber 1 feature. This signature is noticeable from 40 km, in the stratosphere, and extends to 110 km, in the lower thermosphere. Its vertical and longitudinal structure is similar to the one in August but the October feature is significantly



stronger. Starting at 105 km, a zonal wavenumber-4 structure overlaps. This feature is difficult to notice on the $T_{THER}$ fields above 115 km probably because it is weak or/and another component overlaps.

Using the method described in the previous section, we have analyzed MIPAS $\Delta T/2$ fields, analogous to the ones shown in Fig. 1, at each latitude $\phi$ and for the twelve months of the calendar year. For each altitude $z$, we have performed a spectral

analysis to derive the amplitudes $C_k(z,\phi)$ and phases $\theta_k(z,\phi)$ embedding the combination of the tidal modes with daily frequency $n$ and zonal wavenumber $s$ such that $|n_{odd} - s| = k$ (see Table 1).

Figure 2 shows the derived average amplitude spectra during equinox months (April and October) and solstice months (January and July) at 88 and 110 km from wavenumber 0 to 4. The dominance of some modes is evident and depends on season and altitude.

For latitudes within 45° North and South, the $|n_{odd} - s|$=0 mode is the strongest component in the upper mesosphere and below 100 km during April (and also February, March, May and August; not shown). Its amplitude ranges from 9 to 18K. The $|n_{odd} - s|$=1 and $|n_{odd} - s|$=4 oscillations follow in importance in April. During January and October (and also September, November and December; not shown), wavenumber $|n_{odd} - s|$=1 is the dominating component of the tidal field (8-15K) in the mesosphere. In July (and June; not shown), wavenumber $|n_{odd} - s|$=4 is the most important (8K), although $|n_{odd} - s|$=0,1 have

similar amplitudes. MIPAS also detects $|n_{odd} - s|$=0,1 oscillations around 60-70° during the summers with 2K amplitudes.

In the lower thermosphere, around 110 km, the strongest $n - odd$ longitudinal oscillation in MIPAS data is $|n_{odd} - s|$=0 at latitudes smaller than 70° and also $|n_{odd} - s|$=4 at latitudes smaller than 40°(see Fig. 2). This happens throughout the year, the former peaking around 25° N and S (15-20K) (except during June and July) and the latter over the equator (8-16K). In July and August, the $|n_{odd} - s|$=4 oscillation stands out particularly and wavenumber $|n_{odd} - s|$=1 (15K) is then also noticeable at

25°. At highest latitudes, $|n_{odd} - s|$=1 is the only significant mode, although signatures from $|n_{odd} - s|$=2 oscillations are also present, for example, in January (12K).

Next, we describe in detail each of the extracted longitudinal oscillation modes and their seasonal variations. MIPAS Sun-synchronous view provides a fixed local time shot of longitudinal variations. If otherwise not indicated, we use a phase defined in terms of longitude (positive towards the East of Greenwich) of maximum at 10 A.M. The caveat is that, in our case, the phase

is undefined for longitudinally-standing tidal components ($s = 0$).

## 4.1 The $|n_{odd} - s|$=0 mode

The $|n_{odd}-s|$=0 mode, which is just the $\Delta T/2$ zonal mean, contains contribution mainly from the diurnal (DW1) and terdiurnal (TW3) migrating tides. Figures 3-5 include representative latitude-altitude and latitude-time cross sections. We recall that, given MIPAS local time sampling, the values shown correspond to temperature perturbations produced by tides at 10 A.M, including

the information of the local time phase. These do not directly correspond to tidal amplitudes at altitudes where the vertical profiles do not peak.

Temperature perturbation maps at 10 A.M. for DW1 from the Global Scale Wave Model (GSWM) (Hagan and Roble, 2001) at 20-110 km and for DW1 and TW3 from NRLMSISE-00 at 115-150 km are also shown for reference. GSWM results have been convolved considering MIPAS temperature vertical resolution. Note that the contribution from TW3 should be added to





the GSWM model amplitudes for a direct comparison with MIPAS $|n_{odd} - s|$=0. Yue et al. (2013), Pancheva et al. (2013) and Moudden and Forbes (2013) report up to 4K TW3 amplitudes at 90 km in February and September at the equator, and 10K at 110 km at the equator in January and at 50° in the equinoxes and June.

MIPAS $|n_{odd} - s|$=0 latitude-altitude cross-section for October exhibits maximum and minimum values over the equator and at 35° North and South (Fig. 3). The peaks at the equator are located at 42 km (1K), 56 km (-1K), 74 km (5K), 84 km (-11K),
94 km (7K) 104 km (-6K), 110 km (12K) and 120 km (-20K). Out-of-phase peaks (opposite sign) occur at 35° below 105 km. Their altitudes in the Northern Hemisphere (NH) are similar to those at the equator but they are 2 km higher in the Southern Hemisphere. These extra-tropical perturbations are weaker (<1K below 80 km and 2K at 84 km, -6K at 94 km, 4K at 104 km and 15K at 110 km) than in the equator. The oscillation exhibits a decreasing with altitude vertical wavelength of 20-30 km below 110 km, slightly smaller than the values associated with classical tidal theory (Chapman and Lindzen, 1970).

The latitudinal structure of the phase is different at altitudes above 105 km. The perturbations from the tropics to 50°, peaking at 35°(15K), are not out of phase with respect to the equator (they have the same sign at all latitudes). This behavior is consistent with a strong overlapping contribution from TW3, with most pronounced amplitudes at mid-latitudes above 100 km (Moudden and Forbes, 2013).

The sign of the temperature perturbation at 120 km shifts to negative for latitudes between 50°S and 30°N. This behav-
ior could be explained in several ways. The negative values at this altitude accompanying the positive ones at 110 km could come from the same component, TW3. That would imply a vertical wavelength of 20 km, larger than expected from models (Du and Ward, 2010) but similar to those derived from previous measurements in the upper mesosphere (Thayaparan, 1997). Another possibility is a preferential dissipation of the low order DW1 modes from the Hough Mode Extension (HME) (Forbes and Hagan, 1982) at lower altitudes so that higher order modes dominate at these higher altitudes, resulting in a differ-
ent $|n_{odd} - s|$=0 latitudinal distribution. It could also be the consequence of the thermospheric *in situ* DW1, independent to the classical upward propagating DW1 present at altitudes below. That would imply a downward propagation of the thermospheric component, which could be possible under the thermosphere viscid regime (Forbes and Garrett, 1976).

Corresponding GSWM DW1 maxima and minima in October are located within 2-4 km and 1-2K below 100 km, except the maxima at 70 km and 95 km over the equator, which are 5-8K smaller in MIPAS. MIPAS and GSWM DW1 fields above
100 km do not compare that well. The altitude and vertical width of the MIPAS minimum around 105-110 km at the equator are different in the GSWM model. This might be partially explained by the overlapping positive TW3 contribution peaking at 110 km and extending to 50°S-50°N. The measured oscillation at 110 and 115 km is however similar to that centered at 115 km in the NRLMSISE-00 model, being positive at all latitudes. The model DW1 has a slightly longer vertical wavelength on average than that of $|n_{odd} - s|$=0, particularly in the upper mesosphere. This maybe indicates caveats in the representation
of gravity wave and tide interactions in this version of the model (Achatz et al., 2008).

Comparison with 10 A.M.-10 P.M. zonal mean perturbations from the NRLMSISE-00 model, used as *a priori* (see Sect. 2) shows little agreement with MIPAS $|n_{odd} - s|$=0. This strengthens the idea of the *a priori* having small impact on the derived temperature oscillations.




The nearly constant-with-altitude feature above 130 km in Fig. 3 evidences no change in local time phase with altitude. This
is consistent with the effect of the thermospheric *in situ* generated tide. MIPAS measurements occur at LST=10. The phase of
the themospheric *in situ* generated diurnal migrating component occurs at $\varphi_{DW1}$=2-4 P.M., depending on the solar flux input
(Forbes and Garrett, 1976). It is then likely that amplitudes of the thermosphere DW1 tides measured by MIPAS underestimate
total amplitudes, with a factor of attenuation as large as $\cos(n\Omega(10-\varphi_{DW1}))$. MIPAS measures maximum amplitudes of 50K
at 140-150 km in October.

During April (not shown), MIPAS $|n_{odd}-s|$=0 perturbations are 2-5K larger than in October over the equator. Opposite to
results from GSWM DW1, the peak altitudes over the equator and in the NH are slightly higher (2 km) than in October. This
may reflect a change of (local time) phase of the longitudinal oscillation behind. As in October, the strength of the oscillation in
April is latitudinally symmetric, except around 94 km (2K stronger in the NH). MIPAS perturbations in April over the equator
at 84 km and at 35° below 100 km are larger by 5K and 2K, respectively, than in GSWM DW1.

Figure 3 also shows $|n_{odd}-s|$=0 longitudinal perturbations for January. Maximum and minimum values occur at the same
altitudes as in April and are 2-3K smaller than during October below 110 km at low to mid-latitudes. April values are larger
than those of the GSWM model. In the thermosphere, the behavior is similar to that during the rest of the year. Negative
perturbations at 120 km are 5K stronger than in October. Above 130 km, the signature of the *in situ* tide is clear (10-60K),
producing maximum perturbations off the equator and tracking the sub-solar point.

MIPAS sees an oscillation with peaks at 90 and 100 km in January at Southern high latitudes (55-75°S) (Fig. 3). It is in phase
with the perturbations at 35°, probably indicating that it belongs to a different HME mode. Its dependence with season is shown
in the time series at 75°S and 70°N depicted in Fig. 4. A vertical perturbation peaking at 10 A.M. with alternating maxima and
minima at 90, 100 and 110 km is clearly seen from late Spring through Summer at 75°S (2K). This structure is also present at
Northern high latitudes (70°N) at similar altitudes but with slightly weaker amplitudes. This detection corroborates previous
measurements of temperature tides at Southern high latitudes, where 3K amplitude diurnal components with 10 A.M. phases
were measured from ground around 90 km (Lübken et al., 2011). We detect here a Northern hemisphere counterpart.

Latitude-time slices at altitudes where $|n_{odd}-s|$=0 generally peak (84, 110, 120 and 140 km) are shown in Figure 5. The
seasonal and latitudinal variability strongly depends on altitude. Starting with 84 km, the perturbation is stronger during the
equinoxes than during solstices. This could be due to the influence of the seasonal variation of the symmetry of the heating
source (Forbes et al., 2001) but more likely of the zonal winds in the middle atmosphere (McLandress, 2002; Zhu et al., 2005),
probably below 70 km (Achatz et al., 2008). The latitude of the maximum perturbations is slightly shifted towards the equator
during July (as in GSWM DW1; not shown). The inter-hemispheric difference around 35° is generally due to a difference in
the altitude of the peaks (see Sect. 5). There is an isolated minimum (-2K) around 50°N during June-July, with a weaker SH
counterpart, with a significantly smaller amplitude in the model. The amplitude is smaller than 1K for latitudes larger than 70°
at this altitude.

At 110 km, the perturbation is positive all year round, except at all latitudes northern to 70°N. Maxima are generally reached
at 35°, with a 5° shift towards the South in January and towards the North in July (i.e., shifted to higher latitudes in the local
summer), coinciding with previously detected TW3 shifts (Pancheva et al., 2013). Maximum values at these latitudes occur




during the equinoxes, coinciding with the expected maximum contribution from an overlapping DW1. At the equator, opposite
to the GSWM DW1 results, perturbations are positive. The amplitude changes significantly throughout the year (2-16K) with
maximum values in July. Overlapping TW3, which presents significant reduction of amplitude at 110 km right over the equator
during January (Pancheva et al., 2013), is the most likely responsible of this behavior.

At 120 km, MIPAS $|n_{odd} - s|$=0 is negative between 50°S and 50°N. These negative perturbations generally peak at the
same altitude along the year, which implies that the local time phase remains also constant. The latitudinal structure could
correspond to a second-order symmetric mode of DW1 or/and the first TW3 mode. Maximum perturbations occur during the
solstices around 20° South and North (-25K), in contrast to the mesosphere, where maximum amplitudes are found at the
equator during the equinoxes.

At 140 km, the positive peaks, following the sub-solar point, move towards the local summer, when they present maximum
amplitudes. The seasonal and latitudinal behavior of $|n_{odd} - s|$=0 around 120 km presents similarities to that at 140 km, which
could support the possibility of the downward propagation of the *in situ* thermospheric DW1. Negative peaks are measured
during the Autumn around 50° North and South (10-15K). These could be associated to the positive perturbation at lower
altitudes (120 km) at similar latitudes.

## 4.2 The $|n_{odd} - s|$=1 mode

The strongest tidal modes included in the MIPAS $|n_{odd} - s|$=1 longitudinal oscillation are DW2 and D0. TE2 and TW4 os-
cillations are also embedded if present. The diurnal modes are the strongest among them. They probably originate from a
combination of DW1 interaction with sPW1 and the longitudinally varying tropospheric latent heat release related with the
wave-1 surface longitudinal asymmetry (Oberheide et al., 2005).

Figure 6 shows $|n_{odd} - s|$=1 amplitudes and phases derived for November. Amplitudes maximize around the equator and
range from 1-2K from 40 to 65 km, 2-5K from 65 to 75 km and 5-15K from 75 to 110 km. They peak at 86 km. The monotonical
change in phase with altitude indicates a westward propagation. That suggest contribution from DW2. The vertical wavelenght
is 25 km.

$|n_{odd} - s|$=1 presents significant extra-tropical activity, with amplitudes peaking also around 35°S (8K at 90 km and 10K at
105 km) and 35°N (4K at 90 km). The SH amplitudes are thus larger. The phase also increases with altitude towards the West.
The oscillation at these latitudes is out of phase with respect to that at the equator.

A hint of significantly smaller amplitudes (2K at 90 km and 6K 105 km) appears also over 65°S. Those features probably
belong to the same tidal mode because they are in phase and anti-phase with the signals measured at 0°and 35°, respectively,
and also show westward propagation. An analogous signature appears at 65°N but its propagation direction is not as coherent.

It is not clear from MIPAS data how efficiently this oscillation penetrates higher up in the thermosphere. At least, it does
not magnify as altitude increases in the 115-150 km range because the $T_{THER}$ fields for this mode do not show a coherent
structure and amplitudes usually oscillate around temperature error values at these altitudes.

Figure 6 also shows time series of the amplitudes and phases of the MIPAS $|n_{odd} - s|$=1 longitudinal oscillation at 68, 86 and
105 km. The amplitude at 68 km is maximum in July and November (5K) and February (3K). These maxima shift in latitude



from 20°N in January to 20°S in July, i.e., towards the local winter. This off-equator displacement generally occurs at altitudes between 60 and 80 km. It is also present in SABER measurements and is reproduced in CMAM30 (Gan et al., 2014). Its phase displaces from 100°E in January to 200°E in July. Secondary maxima (<2K) appear in July and November 20-30°to the North and to the South of the primary maximum.

The MIPAS $|n_{odd} - s|$=1 wave amplitude at 86 km is maximum at the equator in the equinoxes (10-15K) but also peaks in July (8K). These values are larger than the combination of those reconstructed from ISAMS measurements for DW2 (4K) and D0 (3K) (Forbes and Wu, 2006) or from SABER September 2004 DW2 (2K) and D0 (4K) (Zhang et al., 2006). The typical peaking altitude in MIPAS data below 100 km is 86 km. The phase remains constant throughout the year except in July, when it shifts 180°. This reflects a change in the relative importance of the sources of the underlying tides, which is expected to be stronger in July if tide-wave interaction dominates (see Fig. 3 in Oberheide et al., 2005).

The amplitude of $|n_{odd} - s|$=1 at 86 km reaches 4K from September to December around 35°, it is latitudinally symmetric and out of phase with respect to the equator. This value is in agreement with the combination of DW2 (2K) and D0 (2K) from SABER at a similar altitude in September 2004. The oscillation is also noticed at 35°N in July and 35°S in March (4K).

The time series at 105 km is more structured. Maximum $|n_{odd} - s|$=1 amplitudes occur generally around 30° from November to March, particularly in the SH, and in July, larger in the SH (10K). The phase does not significantly change along the year and is latitudinally asymmetric. This contrasts the behavior at lower altitudes, suggesting main contribution of other HME modes or even tide components. The oscillation at 105 km is significant at the equator from October to December (6-8K) and is out-of-phase with that at 30°S.

These results are consistent with SABER measurements, showing D0 tide maxima at 110 km around 30-40°, particularly during the solstices and also strongest at SH local winter (7K), and DW2 maxima at 100-105 km over the equator, mainly from October to March (7K) (Gan et al., 2014; Truskowski et al., 2014). SABER also showed contribution from DW2 at 35°S but not in the NH, which could also be the responsible for the hemispheric asymmetry here.

Maxima at 105 km, out-of-phase with the ones at 35°S, appear around 65°N and S (6-8K) only during November-December.

### 4.3 The $|n_{odd} - s|$=2 mode

MIPAS $|n_{odd} - s|$=2 longitudinal oscillation embeds the diurnal DE1 and DW3 and the terdiurnal TW1 and TW2 components. The strongest signature among them is the DE1 non-migrating tide and both are most likely originated by non-linear interactions between their migrating counterpart (DW1 and TW3, respectively) and the $s = 1$ stationary planetary wave (SPW1).

Not many tidal analyses report on the detection of these components. Indeed, this mode is weaker than those with $|n_{odd} - s|$=0,1,4 but it is a contribution with a coherent vertical propagation in MIPAS data above 90 km at certain latitudes (see below).

Figure 7 shows latitude-altitude maps of the derived amplitudes and phases for January and September. Both months show eastward propagating oscillations above 90 km at 35° South (2-5K) and North (3-8K in January and 3K in September).

September also exhibits a feature centered at 5-10°S above 75 km that mainly propagates towards the west and peaks at 90 km (6K). It is not clear if the amplitudes measured around 110 km correspond to the same oscillation alone since there is no monotonical phase change with altitude. According to Truskowski et al. (2014), DW3 vertical wavelength prevents its





penetration into the thermosphere, shall it be originated in the troposphere. Therefore, the detection at these high altitudes and the lack of correspondence with an upward propagating signal could be consistent with a local source or with the propagation of a tidal components different to DW3.

The wave spectrum derived from SABER in September around 110 km by Zhang et al. (2006) shows a contribution from DW3 at 10-20°S (6K) and 40-50°S (4K) and small contribution from DE1 around 20°N (<4K). The direction of propagation of the waves derived here only coincide at 35°S. However, CMAM DE1 simulations of Ward et al. (2005) showed a symmetric three maxima latitudinal structure around 80-90 km dominating the mesopause (2K) and turning to an asymmetric two maxima structure in the lower thermosphere. Thus, MIPAS $|n_{odd}-s|$=2 structure probably responds to the overlapping of two significant

contributions, in which DW3 dominates at extra-tropical latitudes and DE1 dominates at the equator.

The time series of $|n_{odd}-s|$=2 at 94 km shows a large seasonal variability. The maximum at 5-10°S appears mainly in May and September with a varying amplitude (bottom left panel in Fig. 7). The oscillation is also noticeable in March-April and September around 30-40°N and S, when the mode is latitudinally asymmetric. July and November reveal a 4K amplitude oscillation closer to the equator (20°) and latitudinally symmetric (in phase).

The time series at 105 km (bottom right panel of Fig. 7) shows an incoherent latitudinal pattern not clearly correlated to the one at 94 km. This suggests contribution of high order Hough modes at higher altitudes, varying along the year. Strong signatures of an anti-symmetric Hough mode are found for January (4-6K), April (2-4K) and July (2K), whereas of a symmetric mode in September.

### 4.4   The $|n_{odd}-s|$=3 mode

The combination of the DE2 and the DW4 tides produces a $|n_{odd}-s|$=3 longitudinal oscillation as seen by MIPAS. DE2 is originated by latent heat release and its latitudinal/seasonal pattern is associated with a modulation by the mean wind (Pancheva et al., 2010).

The $|n_{odd}-s|$=3 mode is one of the dominating MIPAS zonal oscillations over the equator around 110 km in December (12K). Only the $|n_{odd}-s|$=0 mode (migrating tides) is stronger. A latitudinally antisymmetric mode at 90 km dominates MIPAS

temperature amplitudes of $|n_{odd}-s|$=3 during December (see Fig. 8). The amplitudes maximize at tropical latitudes and are larger at 20°S (6K) than at 20°N (4K). The phase dependence with altitude indicates contribution from an eastward propagating wave. The relative importance of the underlying Hough modes of this oscillation changes above 100 km, where a symmetric mode dominates. In that altitude region, amplitudes reach 15K and are significant at latitudes smaller than 30-40°.

Amplitudes from 115-150 km during December also exhibit eastward propagation. Values amplify up to 130-140 km (25K),

where the oscillation starts dissipation. This indicates that, opposite to what Pancheva et al. (2010) state, DE2 may penetrate above 115 km but with a significantly longer vertical wavelength (30 km at 140 km in contrast with 10-12 km in the upper mesosphere). We note that this coherent latitude-altitude behavior at 115-150 km is not found for July. This may be caused by a weaker $|n_{odd}-s|$=3 oscillation then, which MIPAS does not clearly detect.

The change with altitude of the relative importance of the Hough modes in $|n_{odd}-s|$=3 MIPAS mode happens throughout

the year. This is deduced from the different latitudinal distribution of the amplitudes at 90 km and at 110 km (bottom panels in





Fig. 8). Amplitudes at 90 km are maximum during the solstices around 20°. They are larger in the local summer (5-6K) than in the local winter (3-4K). The oscillation also peaks in the solstices at 110 km (12K) and exhibits an off-equator displacement (5-10°) towards the local winter. This result agrees with the DE2 seasonal behavior derived from SABER (Pancheva et al., 2010; Pancheva and Mukhtarov, 2011). MIPAS $|n_{even} - s|$=3 phase around the equator at 110 km moves 20° towards the west

from December to July (not shown), also in agreement with SABER DE2.

### 4.5    The $|n_{odd} - s|$=4 mode

The $|n_{odd} - s|$=4 MIPAS longitudinal variation includes the diurnal non-migrating components DE3 and DW5. These tidal modes are thought to be excited by diurnally-varying latent heat release over the wave-4 land-sea variation. $|n_{odd} - s|$=4 also contains the TW1 and TW2 components. DE3 is the strongest non-migrating tidal effect in the lower thermosphere and has been

widely studied. It is thought to be the main responsible of the wave-4 structure detected at higher altitudes in the thermosphere (Hagan et al., 2007), while the direct absorption of the incoming radiation might play a secondary role (Achatz et al., 2008).

We note here that MIPAS $|n_{odd} - s|$=4 longitudinal oscillation has strong amplitudes in the lower thermosphere (at least below 135 km) over the equator.

Figure 9 shows MIPAS $|n_{odd}-s|$=4 amplitudes and phases altitude-latitude cross-section derived for August. The amplitudes

are significant from the equator to mid-latitudes above 70 km. They increase with altitude, reaching 8K around 95 km, 10K at 100 km, and maximize at 125 km (35K). The mode starts to dissipate at 125 km but keeps consistent upward propagation (monotonically changing phase with altitude). The peak at 125 km is 15 km above the DE3 maximum amplitude modeled by Oberheide and Forbes (2008) for August.

MIPAS $|n_{odd} - s|$=4 generally weakens towards high latitudes. Its amplitude below 120 km is smaller than 5K poleward to

30°. Nevertheless, our results reveal a local maximum in August at 120 km (10K) and 60°. It is not clear if this peak is related to the signals measured at these latitudes at lower altitudes (2K at 90-100 km and 2-4K 110 km).

The $|n_{odd}-s|$=4 phase moves westward as latitude increases, so that the oscillations over the equator and at 20-25° are out of phase. The phase moves eastward as altitude increases, pointing to DE3 as the main contributor. The vertical wavelength in the mesosphere is 8-12 km. The phase plot exhibits a crosswise structure, which softens with height until the lower thermosphere,

where the phase is almost latitudinally symmetric. This indicates the presence of an asymmetric HME mode below 95 km, that is more efficiently dissipated as it propagates upwards than the symmetric ones. This confirms the results for the DE3 HMEs of Oberheide and Forbes (2008). In the lower thermosphere, the vertical wavelength increases to 20 km.

Latitude-time slices at 84, 94,110 and 125 km are plotted in Fig. 10. Maximum amplitudes at 84 km occur around 10-20°. They show up only in the SH in January and July and in the NH in April and September-October. There are North and South

pairs the rest of the year but the NH oscillation is always stronger (7K vs. 4K). Thus, the oscillation tilts to the North during the equinoxes, to the South at the beginning of the NH Winter, and is latitudinally extended and centered over the equator, as a Kelvin wave, during the NH summer. Non-negligible amplitudes appear also at high latitudes, at 60° in July and at 80-85° in November.





The latitudinal tilt at low latitudes at 84 km agrees with findings for DE3 from SABER (Gan et al., 2014; Zhang et al., 2006)
and MLS (Forbes and Wu, 2006) for particular seasons. The 4K larger MIPAS $|n_{odd} - s|$=4 amplitudes are most probably due
to the added contribution from DW5. Zhang et al. (2006) estimated DW5 amplitudes of 2K in September 2004 SABER.

MIPAS $|n_{odd} - s|$=4 oscillation at 94 km shows large amplitudes mainly around the equator (5-8K), consistent with SABER
DE3 (Gan et al., 2014) but 2K larger. Values maximize from June to September and minimize in January. The phase slightly
changes through the year, although generally occurs over Greenwich. The seasonal behavior agrees with a cycle responding
to the combined seasonal variation of the background atmosphere and of the diurnal heat source (Oberheide et al., 2006;
Achatz et al., 2008). Exceptions to this general behavior occur in MIPAS data in November-December, showing peaks at 15°
N and S (5K), and in February, with peaks at 20°N (5K). Expansion of the mode to higher latitudes (50-60°) also occurs in
July-August at this altitude.

The oscillation is significantly amplified at 110 km and 125 km (Fig. 10). The seasonal variability equatorward of 20-30° is
similar to that at lower altitudes. Significant amplitudes are measured from July to October and from March to May (10-15K
at 110 km and 30K at 125 km). MIPAS values at 110 km agree in general with those derived from SABER for DE3 at similar
altitudes (Zhang et al., 2006; Gan et al., 2014), although they are larger (2K) in March. DW5 contribution is also expected at
these altitudes (a maximum of 4K, according to SABER).

This mode also present hemispherically symmetric signatures at 50-60° in July-August at 110 km (2-4K) and 125 km (10K)
and in November at 80° at 110 km (4K).

The phase at 110 km varies over the equator along the year (not shown). It shifts towards the West as latitude increases or,
in terms of local time phase and assuming a dominant eastward propagating component (DE3), it occurs later in the day. The
phase generally presents hemispherical symmetry.

The seasonal behavior reflects, in general at all altitudes, the seasonal change of the relative importance of the different
Hough modes. There is a mayor contribution from a symmetric mode all over the year at thermospheric altitudes, except
from November to January. Then, it is overcome (95-110 km) or competes (around 125 km) with an anti-symmetric mode, in
agreement with HME from Oberheide and Forbes (2008). July also presents a small contribution from the symmetric mode
around 125 km. However, it is then when the asymmetric mode is strongest, whereas Oberheide and Forbes (2008) report it to
be strongest only in January.

**5   Inter-annual variability**

Five years of continuous observations are not enough to unambiguously link atmospheric processes through correlations but
enough to check for tidal quasi-biennial oscillations (QBO). Year to year variability of strengths of the measured longitudinal
oscillations due the effect from the zonal wind Quasi-Biennial-Oscillation is expected (McLandress, 2002; Mayr and Mengel,
2005). Oberheide et al. (2009) and references therein discuss on the possible mechanisms through which the zonal wind QBO
might affect daily temperature oscillations.





At each altitude and latitude, we have decomposed the amplitudes of MIPAS $\Delta T/2$ derived $|n_{odd}-s|$ longitudinal oscillation modes from 2007 to 2012 in six intra-annual sinusoidal components (with 12, 6, 4, 3, 2.4 and 2 month periods), an inter-annual sinusoidal component (with its period as a free parameter) and a component proportional to the solar flux. We have allowed for solar cycle effects, for which we accurately know input energy variation, with the only aim of deriving a more accurate inter-

annual oscillation. Note that MIPAS time coverage spans from solar minimum to solar maximum and, thus, all monotonical variations of temperature amplitudes, like trends, are embedded in this solar component. We have unambiguously found inter-annual variations consistent with a QBO in MIPAS data only for the strong $|n_{odd} - s|$=0 mode. The DE3 component should also present a QBO (Li et al., 2015) but small and it is not surprising that we could no detect it in MIPAS data.

Figure 11 shows typical time series for which we perform the decomposition at each altitude. The examples shown are

for MIPAS derived $|n_{odd} - s|$=0 amplitudes from monthly zonal means from 2007 to 2012 over the equator and at 35°N. In the case of the analysis of the variations of the $|n_{odd} - s|$=0 oscillations with time at a given altitude, we recall that MIPAS $|n_{odd} - s|$=0 values correspond to the amplitudes of a tidal component only at the altitudes for which its phase or anti-phase is 10 A.M., which corresponds to the peaks in the profiles. We already mentioned in Sect. 4.1 that there is a seasonal variation in the altitudes of these peaks, particularly, at extra-tropical latitudes. The altitudes of the $|n_{odd} - s|$=0 maxima vary along a

calendar year as much as 2 km up and down around 78 km and 85 km over the equator but as much as 5 km around 75 km and 3 km at 88 km at 35°N, being higher in the winter (Fig. 11). This also happens in the Southern hemisphere but to a significant lesser extent (not shown). This change of peak altitudes represents a change of phase but not of vertical wavelength because the shift occurs at most altitudes simultaneously. This behavior is repeated every year, pointing to a persistent seasonal effect.

The varying peak altitudes could be originated by a seasonal change of the relative contribution from the different sources

of this component, namely, the semi-annually varying background atmosphere and symmetry of the heating source. Due of its hemispherical asymmetry, we attribute it to the annual oscillation of the background atmosphere, significantly larger in the NH. Interaction with other dynamical features during the winter are not ruled out. This behavior also shows that one should be cautious when interpreting zonal mean intra-annual variability at fixed altitudes from a sun-synchronous instrument data, like MIPAS, because one could be sounding changes of phase of tidal components. Nevertheles, as the change of the altitude of the

peaks at a given month from year to year is not significant, the inter-annual variability can be safely extracted.

A QBO in the $|n_{odd} - s|$=0 mode amplitudes is noticeable in Fig. 11, particularly at low latitudes and even in the middle atmosphere. Figure 12 shows MIPAS $|n_{odd} - s|$=0 amplitudes from 2007 to 2012 at 44, 76 and 86 km over the equator (approximate peaking altitudes in the equinoxes) and the fitted annual, semi-annual and inter-annual components. The inter-annual component (red) competes with the semi-annual oscillation (light blue) below 80 km. We also show two different

representations of the zonal wind stratospheric QBO (SQBO) in Fig. 12: that obtained at the NOAA-ESRL Physical Sciences Division from the zonal average of the 30 mb zonal wind at the equator calculated from the NCEP/NCAR Reanalysis (http://www.esrl.noaa.gov/psd/) and that measured at 30 mb from a radiosonde station in Singapore, compiled by the Freie Universitaet Berlin (http://www.geo.fu-berlin.de/en/met/ag/strat/produkte/qbo/index.html). The period of the inter-annual component (red) of $|n_{odd} - s|$=0 agrees with that of the Singapore winds.



Ekanayake et al. (1997) found that longitudinally propagating tides are generally larger for mean zonal winds blowing in the opposite direction. That implies that DW1 (or TW3), a westward propagating tide, should be larger for westerlies. The phase of the background zonal wind SQBO at 30 mb is opposite to that of the mesospheric QBO (MQBO) (Burrage et al., 1996). Therefore, the effect of SQBO on a tide should be opposite to that of the MQBO.

  MIPAS $|n_{odd} - s|$=0 strengthens at all altitudes simultaneously during the zonal wind SQBO westerly phase (see Fig. 12),

that is, the MQBO easterly phase. That suggest then that the effect on tides is produced by the zonal wind SQBO, although an additional local smaller effect in the mesosphere by the MQBO could also be possible. This supports the argument by (Forbes and Vincent, 1989) of varying stratospheric filtering during upward propagation caused by the varying mean zonal winds instead of a more plausible effect in the mesosphere, where dissipation starts (Oberheide et al., 2009). Note that inter-annual variation at tropospheric altitudes, where the tide mainly originates, can not be ruled out.

A QBO also appears everywhere else where MIPAS $|n_{odd} - s|$=0 mode is significant. Figure 13 shows its QBO derived amplitude. Once more, we recall that MIPAS measures the total QBO effect only at the altitudes where $|n_{odd} - s|$=0 amplitudes are maximum. The QBO oscillation of the $|n_{odd} - s|$=0 mode has amplitudes smaller than 0.5 K amplitudes below 75 km over the equator and below 105 km around 30° North and South, 1 K at 76 km, 2 K at 86 km, 1 K at 98 km and 1.5 K at 110 km. These results agree with those derived from SABER below 90 km (Xu et al., 2009). Above, SABER values are about 1 K larger.

## 6 Summary

The sun-synchronous Michelson Interferometer for Passive Atmospheric Sounding (MIPAS) sounded the atmospheric limb from the stratosphere to the lower thermosphere in a global scale. MIPAS took measurements at two fixed local times during its descending (10 A.M.) and descending (10 P.M.) nodes. Subtraction of the descending and ascending node measurement pairs, $\Delta T/2$, at each latitude, altitude and longitude eliminate the background atmosphere, the persistent (on a daily basis)

longitudinal oscillations (like planetary waves) and also the longitudinal oscillations with daily frequencies such that they are an even integer factor of the 24-hour day (like semi-diurnal tides). Thus, the zonal variation of the $\Delta T/2$ depicts a 'solar-view' mainly of the diurnal and terdiurnal tidal components.

  Extraction of the underlying longitudinal oscillations by spectral analysis isolates the amplitudes and phases of the MIPAS zonal wavenumbers $k$ contributing. Each wavenumber $k$ comprises the combined contribution of the tidal modes fulfilling that

their daily frequency $n$, an odd integer, and their zonal wavenumber $s$, are $|n_{odd} - s|$=$k$. The tidal modes embedded in MIPAS $k$ modes are listed in Table 1.

  MIPAS spectra covered the $CO_2$ 15 $\mu$m and NO 5.3 $\mu$m emissions from which temperatures from 20 to 150 km are derived. We have extracted the wavenumber $k = 0 - 4$ longitudinal waves from these temperatures from April 2007 to March 2012 in a global scale and make them available to the scientific community. To our knowledge, this is the first time temperature zonal

oscillations are derived in this altitude range globally from a single instrument.



We analyze and characterize the behavior of MIPAS $|n_{odd} - s|$ temperature longitudinal oscillations from a climatological point of view from averages of monthly mean MIPAS $\Delta T/2$. The results agree well in general with measurements from other instruments, like SABER or MLS. They reveal that:

- Migrating tidal perturbations with odd $n$ below 105 km are, as expected, stronger during the equinoxes. They are latitudinally symmetric in strength. Their phases exhibit a seasonal variation, with a delay in the winter months that is larger in the Northern hemisphere. The dominating tidal mode at latitudes smaller than $50°$, probably the first mode of the upward propagated DW1, rapidly dissipates above 105 km.

- At 110 km, the major migrating contribution is most likely TW3. Maxima are generally reached at $35°$, with a $5°$ shift to higher latitudes in the local summer. TW3 might also be responsible of the strong migrating perturbations measured at 120 km. Another possibilities are contributions from an upward propagated DW1 high order mode or from the *in situ* generated thermospheric tide.

- The thermospheric DW1 above 130 km produces maximum perturbations off the equator, tracking the sub-solar point and maximizing in the local summer.

- MIPAS measured impact from migrating tides with $n - odd$ at Southern high latitude summer, with alternating maxima and minima at 90 and 100 km in phase with those at $35°$. This agrees with previous ground based detections (Lübken et al., 2011). MIPAS additionally detected a weaker counterpart in the NH summer.

- Besides equatorial, $|n_{odd} - s| = 1$ also exhibits extra-tropical ($35°$) and high-latitude ($65°$) activity in the MLT, particularly in the SH from November to January.

- $|n_{odd} - s| = 2$ presents significant seasonal variability with a latitudinal structure responding to overlapping of two contributions, a westward propagating oscillation that dominates at extra-tropical latitudes and an eastward propagating one that dominates at the equator.

- $|n_{odd} - s| = 3$ is the strongest non-migrating mode in December. An eastward propagating wave, already detected at 70 km, penetrates well in the lower thermosphere with a significantly larger vertical wavelength than in the mesosphere.

- MIPAS shows a prominent wavenumber 4 structure starting at 70 km and maximizing around 135 km (15 km above results from models). The latitudinal distribution reveals contribution from a symmetric Hough mode in the lower thermosphere, which propagates upwards more easily than the antisymmetric one dominating in the mesosphere.

- $|n_{odd} - s| = 4$ expands to higher latitudes in July-August, when hemispherically symmetric footprints are detected at 50-60° above 85 km. Signatures of this mode are also detected in November at $80°$.

We have also studied the inter-annual variability of the amplitudes of the MIPAS $|n_{odd} - s|$ wavenumbers derived from monthly mean MIPAS $\Delta T/2$. We unambiguously detect a $|n_{odd} - s| = 0$ Quasi-Biennial-Oscillation, reaching 2K in the upper




mesosphere at low latitudes. Comparison of tidal QBO and zonal wind stratospheric and mesospheric QBO phases suggests that the effect on tides occurs mainly in the stratosphere and is afterwards propagated upwards.

The good MIPAS temporal resolution and global coverage observations extending from the stratosphere to the lower thermosphere presented here are useful for testing general circulation models considering tidal effects in the MLT region and may well

represent a challenge for them to model their vertical, latitudinal and temporal dependence. A thorough analysis of particular cases found here is needed and will be focus of future work.

*Acknowledgements.* MGC was financially supported by the MINECO under its 'Ramon y Cajal' subprogram. The IAA team was supported by the Spanish MINECO, under project ESP2014-54362-P, and EC FEDER funds. IMK/IAA generated MIPAS data can be accessed after request at https://www.imk-asf.kit.edu/english/1500.php.



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



**Table 1.** Main tidal component contributions resolved in our spectral analysis. The derived amplitudes and phases are a combination of all modes contributing.

| n | wavenumber $\lvert n-s \rvert$ | components |
|---|---|---|
| odd | 0 | DW1+TW3 |
| | 1 | D0+DW2+TW2 |
| | 2 | DW3+DE1+TW1 |
| | 3 | DE2+DW4+T0 |
| | 4 | DE3+DW5+TE1 |





**Figure 1.** Equatorial MIPAS monthly mean temperature difference (left) and anomaly (with respect to the mean value at each altitude; right) of $\Delta T/2$ for August ($1^{st}$ and $2^{nd}$ rows) and October ($3^{rd}$ and $4^{th}$ rows) averaged for 2007-2012. Temperatures below 110 km are retrieved from measurements at 15 $\mu$m and above 115 km from 5.3 $\mu$m. Note the different color scales.







**Figure 2.** MIPAS $\Delta T/2$ average (2007-2012) monthly mean spectra at 88 km (left) and 110 km (right) for January ($1^{st}$ row), April ($2^{nd}$ row), July ($3^{rd}$ row) and October ($4^{th}$ row). MIPAS zonal wavenumbers correspond to $|n_{odd} - s|$ of the daily frequency $n$ and the zonal wavenumber $s$ of comprised tidal modes (see Table 1).





**Figure 3.** Zonal means of $\Delta T/2$ ($|n_{odd} - s|$=0 mode, embedding DW1 and TW3) extracted from average (2007-2012) monthly means of MIPAS $T_{MLT}$ (20-110 km) and $T_{THER}$ (115-150 km) temperatures (left) compared to results for DW1 from GSWM and NRLMSISE-00, respectively (right), for October ($1^{st}$ and $2^{nd}$ rows) and January ($3^{rd}$ and $4^{th}$ rows).





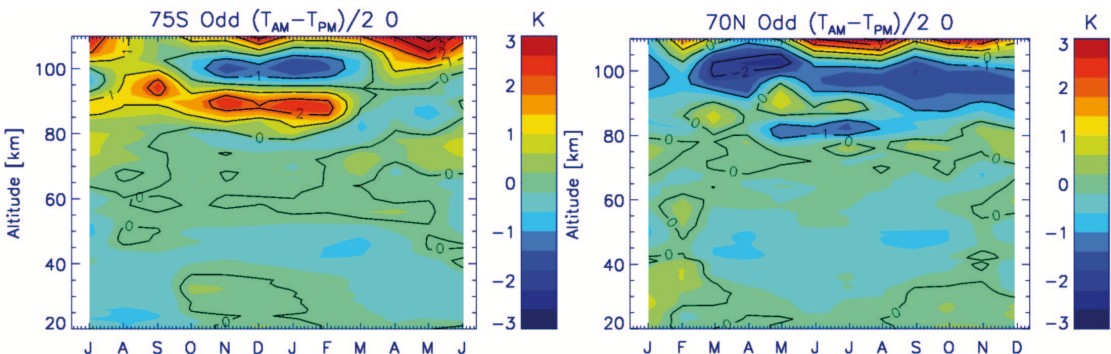

**Figure 4.** Time series of the vertical distribution of average (2007-2012) $\Delta T/2$ monthly zonal means ($|n_{odd} - s|$=0 mode) at high latitudes (75°S, left; 75°N, right).

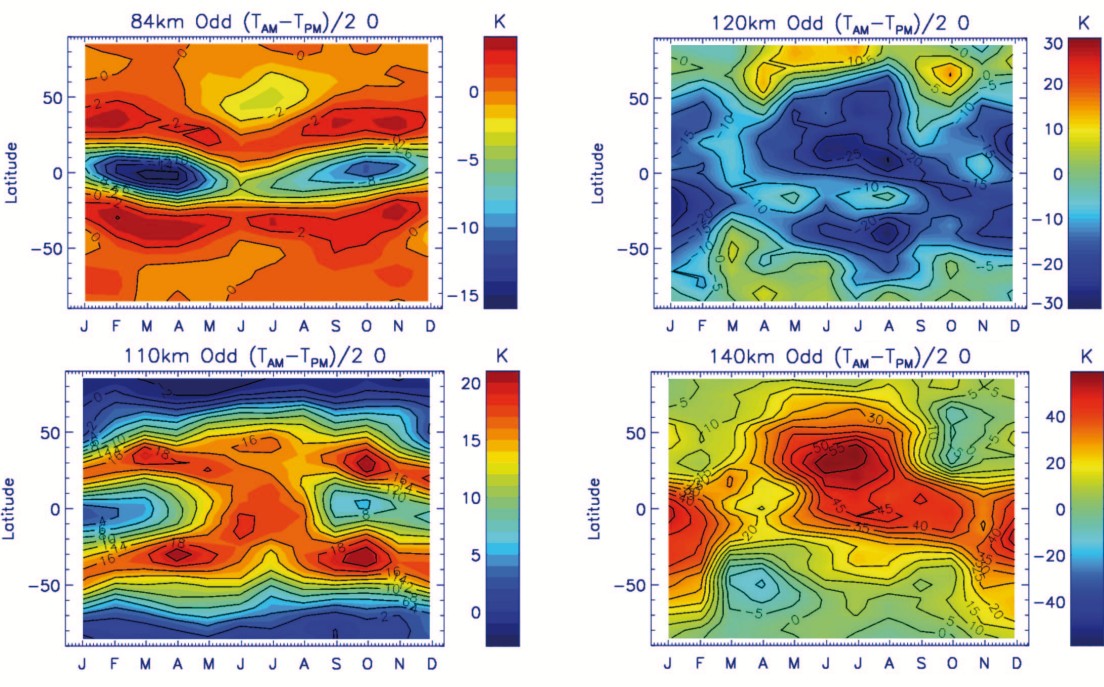

**Figure 5.** Time series of the latitudinal distribution of average (2007-2012) monthly zonal mean $\Delta T/2$ ($|n_{odd} - s|$=0 mode) at 84, 110, 120 and 140 km.





**Figure 6.** Latitude-altitude fields for November ($1^{st}$ row) and latitude-time fields at 68, 86 and 105 km ($2^{nd}$-$4^{th}$ rows) of amplitudes and phases of MIPAS average (2007-2012) $|n_{odd} - s|=1$ mode. It embeds D0, DW2 and TW2 tidal oscillations.




**Figure 7.** Latitude-altitude amplitude (left) and phase (right) fields for January ($1^{st}$ row) and September ($2^{nd}$ row) and latitude-time horizontal slices of amplitudes ($3^{rd}$ row) at 94 km (left) and 105 km (right) of MIPAS average (2007-2012) $|n_{odd} - s|$=2 mode. It comprises effects from the DW3, DE1 and TW1 tidal modes.





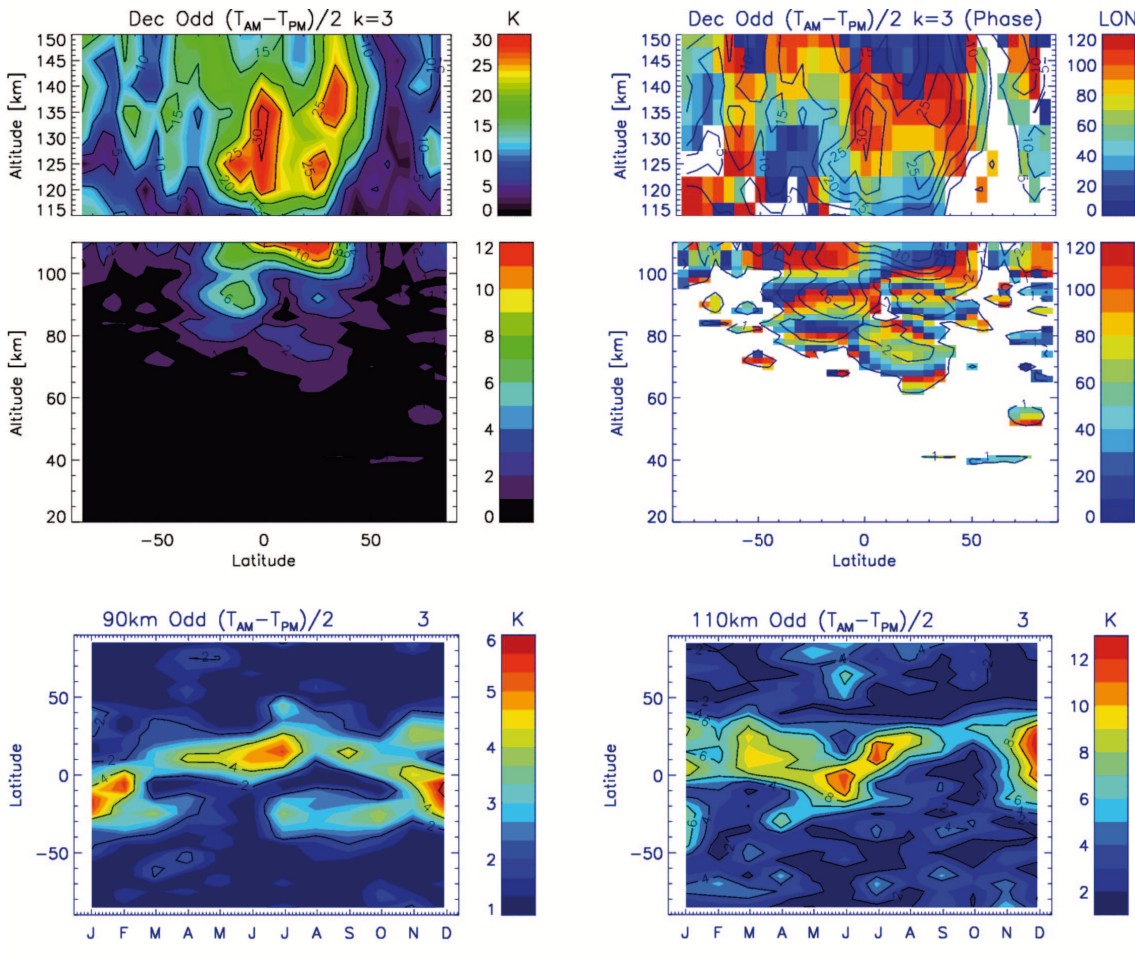

**Figure 8.** Latitude-altitude amplitude (left) and phase (right) fields for December MIPAS average (2007-2012) $|n_{odd} - s|$=3 longitudinal oscillations ($1^{st}$ and $2^{nd}$ row) and average monthly mean time series of $|n_{odd} - s|$=3 amplitudes at 90 km (left) and 110 km (right). They embed contribution from DW3, DE1 and TW1 tides.





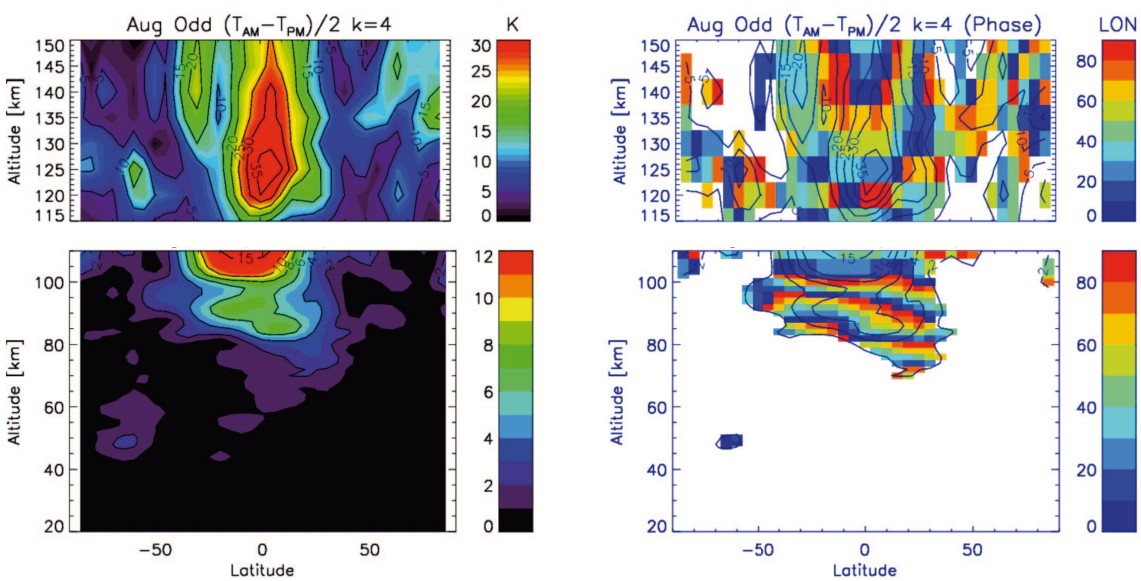

**Figure 9.** August average (2007-2012) monthly mean amplitudes (left) and phases (right) $|n_{odd} - s|$=4 oscillations from MIPAS $T_{THER}$ (115-150 km; upper panels) and $T_{MLT}$ (20-110 km; bottom panels) temperatures. Note the different color scale. They embed the DE3 and TE1 tides.



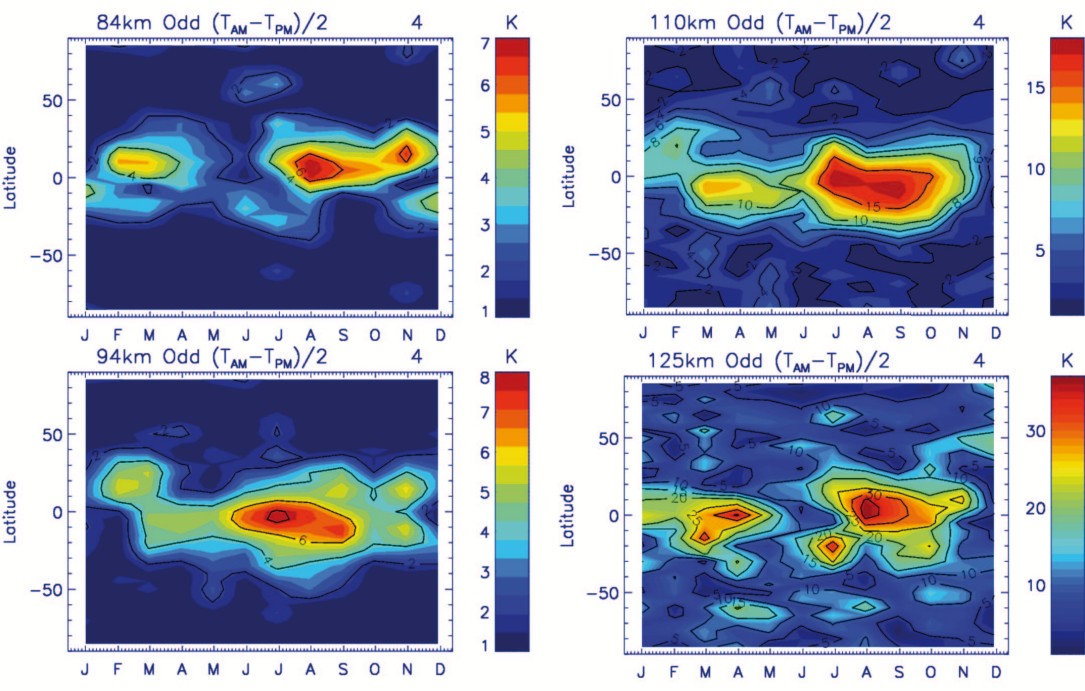

**Figure 10.** Time series of latitudinal distribution of average (2007-2012) monthly means amplitudes for the MIPAS $|n_{odd} - s| = 4$ mode at 84, 94, 110 and 125 km. It embeds DE3 and TE1 tidal oscillations.





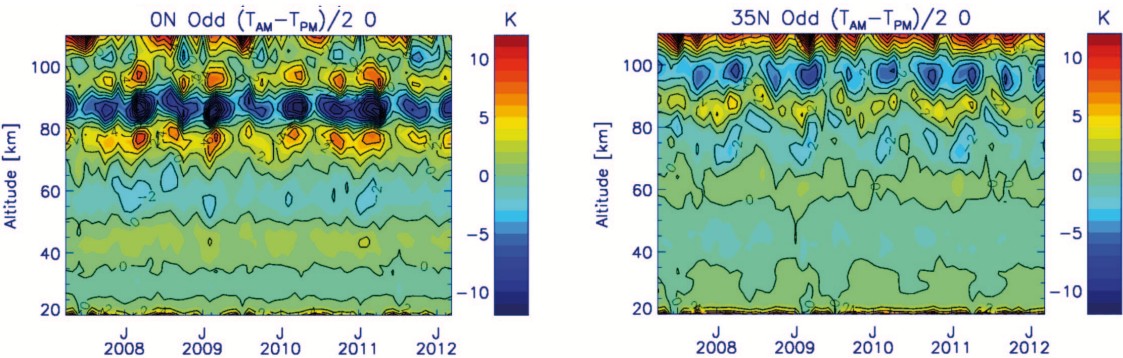

**Figure 11.** Time series of the MIPAS zonal mean of $\Delta T/2$ monthly means ($|n_{odd} - s|$=0 mode) at the equator (left) and over 35°N (right).

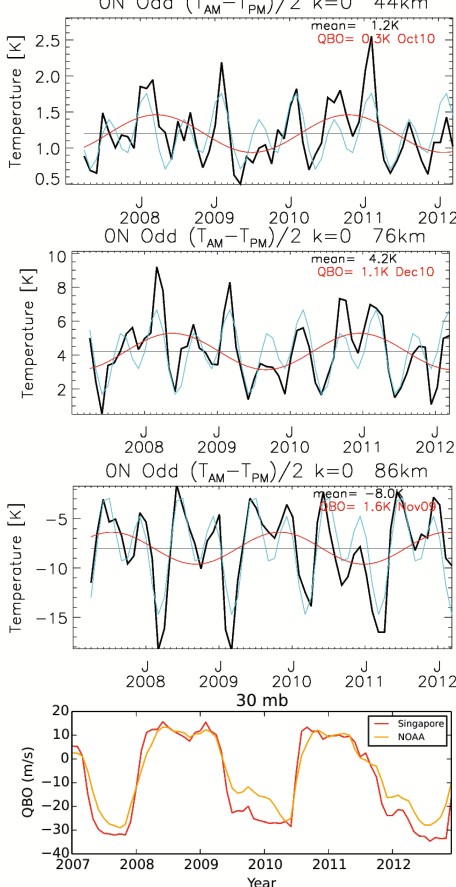

**Figure 12.** MIPAS monthly mean $|n_{odd} - s|$=0 mode perturbations (black lines) at 44, 76 and 86 km and their decomposition in annual plus semi-annual components (blue) and QBO (red). Singapore winds at 30 mb are also plotted in the bottom panel for reference.





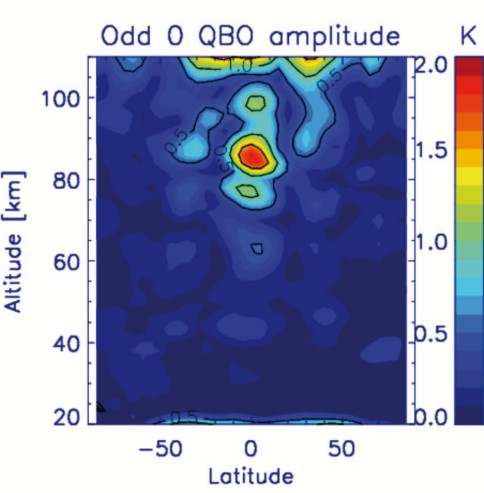

**Figure 13.** QBO amplitudes derived from the MIPAS $|n_{odd} - s|$=0 longitudinal mode extracted from monthly means from April 2007 to March 2012.