# Peer review of "MIPAS observations of longitudinal oscillations in the mesosphere and the lower thermosphere: Climatology of odd-parity daily frequency modes"

_Atmospheric Chemistry and Physics, 2015_

## Referee Comment (RC1) · Anonymous Referee #1 · 7 Mar 2016

This paper presents tidal proxies in MIPAS temperatures inferred as global ascending-descending node differences. The authors explain how these differences highlight diurnal and terdiurnal ("odd") harmonics, and present longitudinal spectral decompositions asa fnction of latitude, altitude and month. The novelty of the work lies in the presentations of MIPAS thermospheric temperatures, the first such results presented above 100 km. This paper is very pertinent to thermospheric dynamics and energetics, and should inform tidal modeling and studies of vertical coupling. I recommend publication after the comments below are addressed, and after a more thorough editing for English grammar.

1. Abstract, line 9-10: The data do not inform you of the QBO transmission process; only the facts should be reported. Change to " ...4) a quasi-biennial oscillation of the migrating tide in the stratosphere and the MLT."

2. Lines 35-36: "Tidal inter-annual variability is thought to be correlated with the El Nino-Southern Oscillation (ENSO)"

The following paper demonstrated clearly how ENSO causes tidal variability.

Lieberman, R. S., D. M. Riggin, D. A. Ortland, S. W. Nesbitt, and R. A. Vincent (2007), Variability of mesospheric diurnal tides and tropospheric diurnal heating during 1997–1998, J. Geophys. Res., 112, D20110, doi:10.1029/2007JD008578.

3. Lines 47-48: "The extent to which tides propagate from the lower atmosphere to the thermosphere or to which changes in lower altitude regions are transmitted by tides to the upper atmosphere or to other latitudes is not completely known."

The following papers should be cited in this section:

Talaat, E. R., and R. S. Lieberman (2010), Direct observations of nonmigrating diurnal tides in the equatorial thermosphere, Geophys. Res. Lett., 37, L04803, doi:10.1029/2009GL041845.

Lieberman, R. S., J. Oberheide, and E. R. Talaat (2013), Nonmigrating diurnal tides observed in global thermospheric winds, J. Geophys. Res. Space Physics, 118, 7384–7397, doi:10.1002/2013JA018975.

Lieberman, R. S., D. M. Riggin, D. A. Ortland, J. Oberheide, and D. E. Siskind (2015), Global observations and modeling of nonmigrating diurnal tides generated by tide-planetary wave interactions, J. Geophys. Res. Atmos., 120, doi:10.1002/2015JD023739.

4. Lines 58-70. This paragraph is extremely confusing, and should be deleted. The authors discuss sun-synchronous sampling before the MIPAS satellite and its orbit are

even mentioned. Non-sun-synchronous sampling is not relevant here.

5. Lines 74-76: "Therefore, they provide a wealth of information on the tidal excitation mechanisms, the processes inducing tidal variability and the lower and upper atmosphere coupling through tides".

Delete this statement. The data cannot accomplish all of that on their own, only in combination with models.

6. Section 3, beginning. I suggest preceeding line 34 with a BRIEF qualitative discussion of the consequences of sun-synchronous sampling. E. g., "Because MIPAS observations occur at 2 fixed local times, migrating tides (that depend only on local time) are seen as invariant features over the course of a day... We explore this and other ramifications of MIPAS sampling in the following discussion..."

7. Line 134: Sentence needs clarification:

"An atmospheric variable X consisting only of tides and a background state at altitude z, latitude, longitude and Universal Time (UT) t′ can be expressed as the sum of the background zonal mean value and the sum of all individual tidal components X′n,s with zonal wavenumber s and wave frequency n at that position and time.."

8. Line 168: Delete "solution".

9. Lines 183-188: Delete.

10. Line 204-205. Change to "an oscillation tilting eastward with height..."

11. Lines 208-209: Change to "Fig. 1 exhibits wave features tilting westward with height (18 km vertical wavelength)..."

12. Lines 211-212: "This feature is difficult to notice..." Delete. In general, do not dwell on things that are not in the data.

13. Line 237: Delete "mainly".

14. Lines 239-240: "including the information of the local time phase." This will confuse the reader, suggest deleting.

15. Line 254: Chapman and Lindzen predicted a vertical wavelength of 27 km for (1,1). Please specify where the observed vertical wavelength od "20-30 km" is smaller or larger than C & L's predictions.

16. Line 305: Change to "...and the zonal winds in the middle atmosphere.."

17. Line 330: Change to "They may originate..." and reference previously mentioned Lieberman et al., 2015. paper.

18. Lines 334-335: "The monotonical change in phase with altitude..." I suggest selecting 3 key latitudes -equatorial, and midlatitude nothern and southern hemisphere - and generating line plots of the phase with altitude. That would make it much easier to see the phase tilt.

19. Lines 340-344: "Those features probably belong..." Delete.

20. Line 372: "MIPAS $|nodd -s|=2$ longitudinal oscillation embeds the diurnal DE1 and DW3 and the terdiurnal TW1 and TW2 components..."

No mention of TW2 is made in Table 1 for MIPAS wave 2, and the math I use (based on Salby's formulas) yields an alias of 1 for TW2.

21. Lines 373-374: "...both are most likely originated by non-linear interactions between their migrating counterpart (DW1 and TW3, respectively) and the s = 1 stationary planetary wave (SPW1)."

The verbiage hereis very muddy. What do you mean by "both"? DE1 and DW3?

What are you proposing for the interactions? DW1 + SPW1, and TW3 + SPW1? Neither one of these produces DE1, DW3, TW1 or TW2.

22. Lines 375-376 ("Not many tidal analyses..."): Delete.

23. Lines 406-407: "The phase dependence with altitude indicates contribution from an eastward propagating wave..." Again, show line plots at representative latitudes. It is actually quite difficult to determine phase tilt in contour plots. Also, the phase appears to be increasing with height in the southern hemisphere, and decreasing with height in the northern hemisphere.

Line 409: "Amplitudes from 115-150 km during December also exhibit eastward propagation." Do you mean to say here that the phases increase with altitude???

24. Line 432: Replace monotonically with "increasing".

25. Lines 437-438: "..phase moves westward as latitude increases... phase moves eastward..." Use phrases such as "phase increases/ decreases with latitude".

26. Lines 487-488: "The DE3 component should also present a QBO (Li et al., 2015) but small and it is not surprising that we could no detect it in MIPAS data." Delete.

27. Lines 520-524: "That suggest then that the effect on tides..." Delete these lines.

28. Figure 13: Convert to a line only at altitude of amplitude maximum.

29. Line 549: Delete "as expected".

30. Lines 576-577: "Comparison of tidal QBO and zonal wind stratospheric..." Delete.

In general the figures were much too small for a review copy.

1. Figure 1: Use monthname-day-year format rather than yyyymmdd in the Figure titles.

2. I suggest either enlarging the lattude-altitude plots, or starting them at z = 60 km. Since the amplitudes are very weak below 60 km, most of these ploots are empty space,and they squish the more interesting behavior at high altitudes into too smallof a space.

---

## Referee Comment (RC2) · Anonymous Referee #3 · 1 Jun 2016

The manuscript aims to diagnose migrating and nonmigrating tides in 5-year monthly mean averages of MIPAS/ENVISAT temperature observations between 20-150 km and 80S-80N. The Sun-synchronous ENVISAT orbit prevents a standard Fourier analysis due to the lacking local solar time coverage. Instead, the manuscript uses the well-known ascending-descending orbit differencing technique to obtain amplitudes and phases of the zonal wavenumber 0-4 patterns in the satellite local solar time frame. The inherent limitation of the approach is that it does not allow one to separate between diurnal and terdiurnal signals, and westward and eastward propagating nonmigrating tidal components. The observed zonally symmetric pattern, that is, the superposition

of the migrating diurnal and terdiurnal tides, is also analyzed on a monthly basis (w/o the 5-year averaging) and compared to the stratospheric Singapore zonal winds, in order to derive a QBO modulation amplitude. Comparisons with the migrating diurnal tide from the GSWM tidal model and NRLMSISE-00 are also shown.

Any new information about tidal characteristics in the 110-150 km region is of value to the aeronomy community since global tidal observations in the transition region into the diffusive regime, where tidal amplitudes and phase are constant with height, are very sparse. As such, I believe the manuscript should ultimately be published. There are, however, a number of important shortcomings in the manuscript that impact its scientific impact.

1. The meat of the manuscript are the data above 110 km since temperature tides in the MLT and below have already been extensively analyzed on monthly mean tides using SABER and MLS data. SABER diagnostics can actually separate tidal components in the MLT and MIPAS does not contribute much here. The bottom line of the lengthy description of MIPAS MLT tidal characteristics in section 4 is that it agrees with SABER. It thus should be scaled back significantly and the paper should focus on the new contribution from MIPAS, that is, tides above 110 km. For example, an interesting finding is the occurrence of the secondary k=4 amplitude maximum above 130 km in Figure 9. This certainly warrants more discussion. I also believe the higher peak altitude of the k=4 pattern warrants more discussion. From a modeling point of view, it is very difficult to shift the maximum towards higher altitudes. This would require a substantial change in the dissipation scheme, resulting in much higher tidal amplitudes in the upper thermosphere. This would then lead to breaking the currently very good agreement with CHAMP and GRACE DE3 tidal diagnostics. In addition, Figure 12 of Lieberman et al. (2013, doi:10.1002/2013JA018975) indicates that the tidal dissipation schemes are actually quite good when comparing to WINDII, including the height of the amplitude maximum. A higher altitude of the DE3 tidal temperature maximum -which would also change the vertical wavelength- would also be difficult to reconcile with DE3

observations above 110 in infrared emissions observed by SABER, since the latter are driven by temperature. See Oberheide et al. (2013, doi:1002/2013JA019278). More discussion of possible reasons for the inconsistency between MIPAS, the current empirical tidal models (and thus also with observed tidal winds from WINDII and infrared emissions from SABER) is needed.

2. There is a considerable number of migrating tide - QBO studies in the MLT from SABER, and it is difficult to see what is new in MIPAS. Everything agrees with SABER. I am OK with leaving section 5 as it is but the earlier work by Huang et al. should be given credit.

3. Tides above 110 km react very strongly to solar conditions, mainly due to the temperature dependence of thermal conductivity. The key figures in the manuscript are 5-year monthly mean averages, from 2007 to 2012, and as such do not account for the important solar cycle dependence. The current results only show that tides are present, but this is something the community already knows. What's needed here is to do the diagnostics for individual years because this would actually help modelers to better constrain dissipative processes and help with our physical understanding of tidal characteristics in the thermosphere.

4. The manuscript does not demonstrate a broad knowledge of previous work in the field. Global tidal observations in the thermosphere are sparse, but the authors seem to be unaware of a number of studies based on WINDII and SABER. See for example See for example Talaat and Lieberman (2010, doi:1029/2009GL041845), Lieberman e tal. (2013, doi:10.1002/2013JA018975), Cho and Shepherd (2015, doi:10.1002/2015JA021903), Oberheide et al. (2013, doi:1002/2013JA019278), and other. I grant that these studies deal with tides in winds and infrared emissions but they have been conclusively connected to in-situ tidal temperature diagnostics from CHAMP and GRACE in the upper thermosphere (see the various papers by Jeff Forbes) using empirical tidal modeling, including the abovementioned solar cycle dependence. I also believe the presented results need to be put more carefully into the context of recent

progress in whole atmosphere modeling, e.g., using WACCM-X, WAM, and GAIA. The current discussion in the GCM context is essentially limited to a one year long run of the CMAM model that has been done a few years ago. CMAM development has been stopped a few years ago and more up-to-date models (or at the very least the more recent eCMAM30 run) are more appropriate for this discussion.

5. What is the purpose of the GSWM/MSIS comparisons? What model version has been used and how? The given GSWM reference points to an old TIME-GCM study (where GSWM was used as a lower boundary condition only). There are several versions of GSWM around, the most recent one is GSWM-09 (see papers by Xiaoli Zhang). I doubt that this one has been used since no reference is given. Older GSWM versions had issues with seasonal variations and partly did not include the in-situ tidal forcing in the thermosphere. Also, GSWM is for 110 sfu (if I remember correctly) and does not include any solar flux dependence. I am also puzzled to see that MSIS shows such a poor agreement with MIPAS. The MSIS amplitudes close to 150 km look way too small for migrating tides. Forbes et al. (2011, doi:10.1029/2011JA016855) compare the MSIS migrating diurnal tide at 400 km with CHAMP and GRACE. The agreement is actually quite good with amplitudes on the order of 120 K.

6. Several conclusions are not supported by the data and speculation. (1) How do you know the propagation direction from the latitude/height Figure 7 (section 4.3, section 6)? Longitude/height plots give some indication about propagation direction, assuming that all tidal signals are propagating upward w/o any possible downward propagation or in-situ forcing (which is an assumption that needs to be stated!). (2) The TW3 as the leading migrating component at 110 km (section 4.1, section 6) is mere speculation since MIPAS cannot separate DW1 and TW3. In-situ DW1 forcing is as likely (or more likely).

7. Methodology section 3. I doubt that a non-expert in tidal satellite diagnostics will understand this section. It gives an overly complicated description of a well-established method that has been applied over the past 20 years to every single remote sensing

infrared instrument when looking into tides. I strongly suggest to significantly shorten the section (or moving the shortened version into section 2 altogether). If the authors insist to keep this level of detail, the section should be moved into an appendix, but with the addition of a few intermediate steps that have been omitted, to help readers not familiar with the satellite orbit geometries and sampling.

Specific comments.

line 523. Oberheide et al. (2009) do not discuss the QBO in the westward propagating migrating tide, only in the eastward propagating DE3.

The lower altitude in the Figures should be moved up to 50 or 70 km. There's not much tidal activity going on in the stratosphere.

The language is mostly fine but another round of proof-reading by the native speaker on the co-author list would be good.
* * *

---

## Author Comment (AC1) · 26 Jul 2016

**Responses to Anonymous Referee #1**

• **Reviewer's comment:**
*This paper presents tidal proxies in MIPAS temperatures inferred as global ascending-descending node differences. The authors explain how these differences highlight diurnal and terdiurnal ("odd") harmonics, and present longitudinal spectral decompositions asa fnction of latitude, altitude and month. The novelty of the work lies in the presentations of MIPAS thermospheric temperatures, the first such results presented above 100 km. This paper is very pertinent to thermospheric dynamics and energetics, and should inform tidal modeling and studies of vertical coupling. I recommend publication after the comments below are addressed, and after a more thorough editing for English grammar.*

**Author's response:**
We thank the Reviewer for his/her comments. We think we addressed them all and they certainly improved the manuscript. A native English speaker has checked the grammar of the reviewed version.

Main changes of the manuscript are: update of the version of retrieved thermospheric temperatures (results barely change); inclusion of new figures with lower altitude of 40 km; old Sect. 3 has been moved to an Appendix; the discussion on thermospheric tides has been extended.

• **Reviewer's comment:**
1. Abstract, line 9-10: The data do not inform you of the QBO transmission process; only the facts should be reported. Change to " ...4) a quasi-biennial oscillation of the migrating tide in the stratosphere and the MLT."

**Author's response:**
Done.

• **Reviewer's comment:**
*2. Lines 35-36: "Tidal inter-annual variability is thought to be correlated with the El Nino-Southern Oscillation (ENSO)"*
*The following paper demonstrated clearly how ENSO causes tidal variability.*
*Lieberman, R. S., D. M. Riggin, D. A. Ortland, S. W. Nesbitt, and R. A. Vincent (2007), Variability of mesospheric diurnal tides and tropospheric diurnal heating during 1997–1998, J. Geophys. Res., 112, D20110, doi:10.1029/2007JD008578.*

**Author's response:**
We deleted 'thought to be' and included the reference.

• **Reviewer's comment:**
*3. Lines 47-48: "The extent to which tides propagate from the lower atmosphere to the thermosphere or to which changes in lower altitude regions are transmitted by tides to the upper atmosphere or to other latitudes is not completely known."*
*The following papers should be cited in this section:*
*Talaat, E. R., and R. S. Lieberman (2010), Direct observations of nonmigrating diurnal tides in the equatorial thermosphere, Geophys. Res. Lett., 37, L04803, doi:10.1029/2009GL041845.*
*Lieberman, R. S., J. Oberheide, and E. R. Talaat (2013), Nonmigrating diurnal tides observed in global thermospheric winds, J. Geophys. Res. Space Physics, 118, 7384– 7397, doi:10.1002/2013JA018975.*

*Lieberman, R. S., D. M. Riggin, D. A. Ortland, J. Oberheide, and D. E. Siskind (2015), Global observations and modeling of nonmigrating diurnal tides generated by tide-planetary wave interactions, J. Geophys. Res. Atmos., 120, doi:10.1002/2015JD023739.*

**Author's response:**

Done. They are now included in the next paragraph of the manuscript.

• **Reviewer's comment:**

*4. Lines 58-70. This paragraph is extremely confusing, and should be deleted. The authors discuss sun-synchronous sampling before the MIPAS satellite and its orbit are even mentioned. Non-sun-synchronous sampling is not relevant here.*

**Author's response:**

An unavoidable caveat in sun-synchronous measurements is aliasing of certain tidal modes. Slowly precessing satellites eventually provide a complete local time coverage that overcomes that problem but at the expense of the temporal resolution. We did not delete the whole paragraph in the text but we significantly shorten and clarify it. We have also exchanged the order of this paragraph and the (old) next one.

• **Reviewer's comment:**

*5. Lines 74-76: "Therefore, they provide a wealth of information on the tidal excitation mechanisms, the processes inducing tidal variability and the lower and upper atmosphere coupling through tides". Delete this statement. The data cannot accomplish all of that on their own, only in combination with models.*

**Author's response:**

We think that the study of correlations among measured longitudinal oscillations and also with external sources help to understand excitation mechanisms (Immel et al., 2006; Xu et al., 2014), atmospheric coupling (Oberheide and Forbes, 2008; Forbes et al., 2009) and variability processes (Zhang and Shepherd, 2005; Forbes et al, 2008). Nevertheless, we agree that this information must be demonstrated in combination with models. Therefore, we softened the sentence by substituting 'provide a wealth of information' by 'may reveal indications'.

• **Reviewer's comment:**

*6. Section 3, beginning. I suggest preceeding line 134 with a BRIEF qualitative discussion of the consequences of sun-synchronous sampling. E. g., "Because MIPAS observations occur at 2 fixed local times, migrating tides (that depend only on local time) are seen as invariant features over the course of a day... We explore this and other ramifications of MIPAS sampling in the following discussion..."*

**Author's response:**

Following the suggestion from the other referee, we have moved Section 3 to an Appendix. The new Appendix has been also modified. Nevertheless, we included the sentence suggested by the referee in Sect. 3.1.

• **Reviewer's comment:**

*7. Line 134: Sentence needs clarification: "An atmospheric variable X consisting only of tides and a background state at altitude z, latitude, longitude and Universal Time (UT) tâAˇš can be expressed as the sum of the background zonal mean value and the sum of all individual tidal components Xn,s with zonal wavenumber s and wave frequency n at that position and time.."*

**Author's response:**

We have slightly changed the sentence and shortened the paragraph.

- **Reviewer's comment:**
  *8. Line 168: Delete "solution".*
  **Author's response:**
  Done.

- **Reviewer's comment:**
  *9. Lines 183-188: Delete.*
  **Author's response:**
  We have almost deleted the complete paragraph but have left the reference.

- **Reviewer's comment:**
  *10. Line 204-205. Change to "an oscillation tilting eastward with height..."*
  **Author's response:**
  Done.

- **Reviewer's comment:**
  *11. Lines 208-209: Change to "Fig. 1 exhibits wave features tilting westward with height (18 km vertical wavelength)..."*
  **Author's response:**
  Done.

- **Reviewer's comment:**
  *12. Lines 211-212: "This feature is difficult to notice..." Delete. In general, do not dwell on things that are not in the data.*
  **Author's response:**
  It is true that this is not a striking feature in Fig. 1 in October but, as it is shown later (Fig.10), it certainly is in the data. It is weak and masked by other oscillations in Fig. 1. We substituted 'difficult to notice' by 'perceived (...) but it is not as evident as at lower altitudes or in August,'.

- **Reviewer's comment:**
  *13. Line 237: Delete "mainly".*
  **Author's response:**
  Done.

- **Reviewer's comment:**
  *14. Lines 239-240: "including the information of the local time phase." This will confuse the reader, suggest deleting.*
  **Author's response:**
  Done. (We moved the sentence suggested by the reviewer some comments above to this paragraph).

- **Reviewer's comment:**
  *15. Line 254: Chapman and Lindzen predicted a vertical wavelength of 27 km for (1,1). Please specify where the observed vertical wavelength of "20-30 km" is smaller or larger than C & L's predictions.*
  **Author's response:**

Done.

- **Reviewer's comment:**
*16. Line 305: Change to "...and the zonal winds in the middle atmosphere.."*
**Author's response:**
Done.

- **Reviewer's comment:**
*17. Line 330: Change to "They may originate..." and reference previously mentioned Lieberman et al., 2015. paper.*
**Author's response:**
Done.

- **Reviewer's comment:**
*18. Lines 334-335: "The monotonical change in phase with altitude..." I suggest selecting 3 key latitudes -equatorial, and midlatitude nothern and southern hemisphere - and generating line plots of the phase with altitude. That would make it much easier to see the phase tilt.*
**Author's response:**
This is a great suggestion. We now include two new figures: new Figs. 7 and 10.

- **Reviewer's comment:**
*19. Lines 340-344: "Those features probably belong..." Delete.*
**Author's response:**
Done.

- **Reviewer's comment:**
*20. Line 372: "MIPAS |nodd −s|=2 longitudinal oscillation embeds the diurnal DE1 and DW3 and the terdiurnal TW1 and TW2 components..."*
*No mention of TW2 is made in Table 1 for MIPAS wave 2, and the math I use (based on Salby's formulas) yields an alias of 1 for TW2.*
**Author's response:**
As correctly pointed out by the referee, TW2 yields 1, as shown in Table 1. TW2 deleted in the text.

- **Reviewer's comment:**
*21. Lines 373-374: "...both are most likely originated by non-linear interactions between their migrating counterpart (DW1 and TW3, respectively) and the s = 1 stationary planetary wave (SPW1)."*
*The verbiage here is very muddy. What do you mean by "both"? DE1 and DW3?*
*What are you proposing for the interactions? DW1 + SPW1, and TW3 + SPW1? Neither one of these produces DE1, DW3, TW1 or TW2.*
**Author's response:**
That sentence was definitely wrong. These non-migrating tides could originate from non-linear interactions between their migrating counterparts (DW1 and TW3) and s=2 stationary planetary waves (SPW2). Changed.

- **Reviewer's comment:**
*22. Lines 375-376 ("Not many tidal analyses..."): Delete.*

**Author's response:**
Re-written.

• **Reviewer's comment:**
*23. Lines 406-407: "The phase dependence with altitude indicates contribution from an eastward propagating wave..." Again, show line plots at representative latitudes. It is actually quite difficult to determine phase tilt in contour plots. Also, the phase appears to be increasing with height in the southern hemisphere, and decreasing with height in the northern hemisphere.*
**Author's response:**
Thank you again for this suggestion. As mentioned above, we now include Fig. 10 showing the phase tilt with height.

• **Reviewer's comment:**
*Line 409: "Amplitudes from 115-150 km during December also exhibit eastward propagation."*
*Do you mean to say here that the phases increase with altitude???*
**Author's response:**
Yes, we did mean that phases increase with altitude. Re-written.

• **Reviewer's comment:**
*24. Line 432: Replace monotonically with "increasing".*
**Author's response:**
Done. Sentence re-written.

• **Reviewer's comment:**
*25. Lines 437-438: "..phase moves westward as latitude increases... phase moves eastward..." Use phrases such as "phase increases/ decreases with latitude".*
**Author's response:**
Done.

• **Reviewer's comment:**
*26. Lines 487-488: "The DE3 component should also present a QBO (Li et al., 2015) but small and it is not surprising that we could no detect it in MIPAS data." Delete.*
**Author's response:**
We re-phrased the sentence.

• **Reviewer's comment:**
27. Lines 520-524: "That suggest then that the effect on tides..." Delete these lines.
**Author's response:**
We think that, assuming the argument from Ekanayake et al., (1997) (stronger tides when direction of propagation is opposite to the wind), if the QBO effect on the migrating tide (westward propagation) were produced locally in the mesosphere, the tide amplitude would strengthen when the zonal wind mesospheric QBO were in its westerly phase (wind towards the east). That is *not* the case in MIPAS data. The tide, on the contrary, strengthens during the westerly phase of the stratospheric QBO. Therefore, it is *more likely* that the tide QBO originates in the stratosphere. The later is however not necessarily true and that is why we leave a door open and write in Sect. 4 (old Sect. 5) that other mechanisms cannot be ruled out.

We think this reasoning is not a speculation but suggests that the mesospheric tide QBO is not a local effect.

Nevertheless, we re-wrote the paragraph and say that MIPAS suggests that a stratospheric effect is more likely than a mesospheric effect. We hope that the argument is clearer now and eludes speculations.

- **Reviewer's comment:**
*28. Figure 13: Convert to a line only at altitude of amplitude maximum.*
**Author's response:**
We think that substituting this color map by a 1D plot with amplitude vs. latitude at only one selected altitude would eliminate a lot of information. We believe the figure is simple and clear enough. We think that all information on the dependence of the amplitude on altitude and latitude can easily be shown with this map. Additionally, Fig. 14 (old Fig.12) gives some more information at the latitude where the QBO peaks.

- **Reviewer's comment:**
*29. Line 549: Delete "as expected".*
**Author's response:**
Done

- **Reviewer's comment:**
*30. Lines 576-577: "Comparison of tidal QBO and zonal wind stratospheric..." Delete.*
**Author's response:**
Changed to: ' the effect on tides does not mainly occur in the mesosphere' (see argument in response to comment 27).

- **Reviewer's comment:**
*In general the figures were much too small for a review copy.*
**Author's response:**
We re-did all figures. We hope they meet the standards of ACP.

- **Reviewer's comment:**
*1. Figure 1: Use monthname-day-year format rather than yyyymmdd in the Figure titles.*
**Author's response:**
Done.

- **Reviewer's comment:**
*2. I suggest either enlarging the latitude-altitude plots, or starting them at z = 60 km. Since the amplitudes are very weak below 60 km, most of these ploots are empty space,and they squish the more interesting behavior at high altitudes into too small of a space.*
**Author's response:**
We moved the Figures up to 40 km. We note that Zeng et al. (2008) detected DW1 activity (although with very small amplitudes 1K) already in the lower stratosphere.

**References (used in this document but not included in the manuscript)**

Forbes, J.; M.; Zhang, X.; Palo, S.; Russell, J.; Mertens, C.; J. & Mlynczak, M. Tidal variability in the ionospheric dynamo region, *Journal of Geophysical Research,* **2008***, 113*, A02310 .

Oberheide, J.; Forbes, J.M. Tidal propagation of deep tropical cloud signatures into the thermosphere from TIMED observations, *Geophys. Res. Lett.,* **2008***, 35*, L04816.

Xu, J.; Smith, A. K.; Liu, M.; Liu, X.; Gao, H.; Jiang, G.; Yuan, W. Evidence for nonmigrating tides produced by the interaction between tides and stationary planetary waves in the stratosphere and lower mesosphere, *J. Geophys. Res.,* **2014***, 119*, 471-489.

Zhang, S.; P.; Shepherd, G.  G. Variations of the mean winds and diurnal tides in the mesosphere and lower thermosphere observed by WINDII from 1992 to 1996 *Geophys. Res. Lett.,* **2005***, 32*, L14111.

Zeng, Z., W. Randel, S. Sokolovskiy, C. Deser, Y.-H. Kuo, M. Hagan, J. Du, and W. Ward (2008), Detection of migrating diurnal tide in the tropical upper troposphere and lower stratosphere using the Challenging Minisatellite Payload radio occultation data, J. Geophys. Res., 113, D03102, doi:10.1029/2007JD008725.

---

## Author Comment (AC2) · 26 Jul 2016

• **Reviewer's comment:**

*The manuscript aims to diagnose migrating and nonmigrating tides in 5-year monthly mean averages of MIPAS/ENVISAT temperature observations between 20-150 km and 80S-80N. The Sun-synchronous ENVISAT orbit prevents a standard Fourier analysis due to the lacking local solar time coverage. Instead, the manuscript uses the well-known ascending-descending orbit differencing technique to obtain amplitudes and phases of the zonal wavenumber 0-4 patterns in the satellite local solar time frame. The inherent limitation of the approach is that it does not allow one to separate between diurnal and terdiurnal signals, and westward and eastward propagating nonmigrating tidal components. The observed zonally symmetric pattern, that is, the superposition of the migrating diurnal and terdiurnal tides, is also analyzed on a monthly basis (w/o the 5-year averaging) and compared to the stratospheric Singapore zonal winds, in order to derive a QBO modulation amplitude. Comparisons with the migrating diurnal tide from the GSWM tidal model and NRLMSISE-00 are also shown.*

*Any new information about tidal characteristics in the 110-150 km region is of value to the aeronomy community since global tidal observations in the transition region into the diffusive regime, where tidal amplitudes and phase are constant with height, are very sparse. As such, I believe the manuscript should ultimately be published. There are, however, a number of important shortcomings in the manuscript that impact its scientific impact.*

**Author's response:**

We thank the referee for his/her very useful comments that we think have improved the manuscript. We have taken into account his/her suggestions.

Main changes of the manuscript are: update of the version of retrieved thermospheric temperatures (results barely change); inclusion of new figures with lower altitude of 40 km; old Sect. 3 has been moved to an Appendix; introduction and discussion on thermospheric tides has been extended.

• **Reviewer's comment:**

*1. The meat of the manuscript are the data above 110 km since temperature tides in the MLT and below have already been extensively analyzed on monthly mean tides using SABER and MLS data. SABER diagnostics can actually separate tidal components in the MLT and MIPAS does not contribute much here. The bottom line of the lengthy description of MIPAS MLT tidal characteristics in section 4 is that it agrees with SABER. It thus should be scaled back significantly and the paper should focus on the new contribution from MIPAS, that is, tides above 110 km.*

**Author's response:**

MIPAS, MLS and SABER are different instruments on different platforms. We think measurements of all three of them (and other instruments) are equally interesting and, thus, it is worth reporting them all.

MIPAS and SABER results qualitatively agree, a result itself, but they do not coincide. As explained several times in the manuscript, temporal resolution of SABER standard analyses is worse than that of MIPAS (2 months vs. 1 month).

Opposite to SABER, that yaws every two months to observe the two poles alternatively, MIPAS provides a pole-to-pole view of the MLT. In this context, the effect of the mesospheric migrating tide measured at high latitudes simultaneously in both hemispheres can be reported here. Also, high latitude tide activity can be tracked along the year by MIPAS (k=1,4).

Compared to MLS, MIPAS vertical resolution, that affects wave structures (see Sect. 2), is better.

Finally, MIPAS offers the rare opportunity to observe the atmosphere from the stratosphere up to 150 km globally. Since tides generally propagate from low altitudes, a continuous vertical coverage from a single instrument is an advantage, as we mention in the abstract and in Sect. 1.

Nevertheless, following the referee's suggestion, we tried to shorten the text deleting several full paragraphs on the discussion below 110km in Sect. 3, particularly, in its introduction and Sections 3.1 and 3.2 but also in Sect. 3.5.

- **Reviewer's comment:**
*For example, an interesting finding is the occurrence of the secondary k=4 amplitude maximum above 130 km in Figure 9. This certainly warrants more discussion. I also believe the higher peak altitude of the k=4 pattern warrants more discussion. From a modeling point of view, it is very difficult to shift the maximum towards higher altitudes. This would require a substantial change in the dissipation scheme, resulting in much higher tidal amplitudes in the upper thermosphere. This would then lead to breaking the currently very good agreement with CHAMP and GRACE DE3 tidal diagnostics. In addition, Figure 12 of Lieberman et al. (2013, doi:10.1002/2013JA018975) indicates that the tidal dissipation schemes are actually quite good when comparing to WINDII, including the height of the amplitude maximum. A higher altitude of the DE3 tidal temperature maximum -which would also change the vertical wavelength- would also be difficult to reconcile with DE3 observations above 110 in infrared emissions observed by SABER, since the latter are driven by temperature. See Oberheide et al. (2013, doi:1002/2013JA019278). More discussion of possible reasons for the inconsistency between MIPAS, the current empirical tidal models (and thus also with observed tidal winds from WINDII and infrared emissions from SABER) is needed.*

**Author's response:**
We appreciate this comment. We do not actually see such a disagreement with models, as the referee mentions. Amplitudes over the equator increase with altitude from the upper mesosphere to 120-125 km, where they reach its maximum in the altitude range examined in this work. This qualitatively agrees with the results for temperature from models, that place de DE3 peak around 110-115 km (see Sect. 3.5 for references). The 10 km shift might be partially explained by the large vertical resolution of MIPAS temperatures in the thermosphere. We note that, as pointed by the referee, the peak in the u and v fields are placed around 105 km. We have included a broader discussion on the k=4 peak altitude in Sect. 3.5.

- **Reviewer's comment:**
*2. There is a considerable number of migrating tide - QBO studies in the MLT from SABER, and it is difficult to see what is new in MIPAS. Everything agrees with SABER. I am OK with leaving section 5 as it is but the earlier work by Huang et al. should be given credit.*

**Author's response:**
Done.

- **Reviewer's comment:**
*3. Tides above 110 km react very strongly to solar conditions, mainly due to the temperature dependence of thermal conductivity. The key figures in the manuscript are 5-year monthly mean averages, from 2007 to 2012, and as such do not account for the*

*important solar cycle dependence. The current results only show that tides are present, but this is something the community already knows. What's needed here is to do the diagnostics for individual years because this would actually help modelers to better constrain dissipative processes and help with our physical understanding of tidal characteristics in the thermosphere.*

**Author's response:**

We agree that analysis of individual years providing information on influence of solar conditions would be of great interest. There are several reasons why we think it is not adequate to perform such analysis from MIPAS data. Firstly, continuous observations should extend for a big portion of the solar cycle or, at least, for a portion for which changes in solar flux are important. Unfortunately, that is not the case for MIPAS data. The data cover solar flux changes of the order of 50 sfu. Maximum variations from 2007 to 2012 of monthly 10AM DW1 temperature contributions at 150km in NRLMSISE-00 are 4-7K (depending on season). Secondly, not only DW1 amplitudes vary with the solar cycle but also phases and those would be tracked in MIPAS locked LT measurements together with the solar cycle impact. Thirdly, regarding other significant modes in the thermosphere, variations along the solar cycle in the altitude range studied here are expected to be hardly detectable (see Fig.4 for DE3 in Oberheide et al., 2009)

We note that, even if the community already knows that the tides are present, a quantitative analysis, even from averages, is valuable. We focus our comparisons with SABER because it has provided reliable measurements of tides and they have been used as input for several models. We additionally recall that temperature tide measurements in the E-region are scarce and those covering from the stratosphere up to that region from a single instrument inexistent.

- **Reviewer's comment:**

*4. The manuscript does not demonstrate a broad knowledge of previous work in the field. Global tidal observations in the thermosphere are sparse, but the authors seem to be unaware of a number of studies based on WINDII and SABER. See for example See for example Talaat and Lieberman (2010, doi:1029/2009GL041845), Lieberman e tal. (2013, doi:10.1002/2013JA018975), Cho and Shepherd (2015, doi:10.1002/2015JA021903), Oberheide et al. (2013, doi:1002/2013JA019278), and other. I grant that these studies deal with tides in winds and infrared emissions but they have been conclusively connected to in-situ tidal temperature diagnostics from CHAMP and GRACE in the upper thermosphere (see the various papers by Jeff Forbes) using empirical tidal modeling, including the abovementioned solar cycle dependence.*

**Author's response:**

We have included a number of new references along the text and tried to put our results into their context (particularly in the introduction but also along Section 3).

- **Reviewer's comment:**

*I also believe the presented results need to be put more carefully into the context of recent progress in whole atmosphere modeling, e.g., using WACCM-X, WAM, and GAIA. The current discussion in the GCM context is essentially limited to a one year long run of the CMAM model that has been done a few years ago. CMAM development has been stopped a few years ago and more up-to-date models (or at the very least the more recent eCMAM30 run) are more appropriate for this discussion.*

**Author's response:**

The aim of this paper is to report and describe MIPAS measurements of tides in the context of previous measurements but not to perform a thorough comparison with models. That will be the focus of future work, for which taking into account MIPAS sampling and vertical resolution is needed. Nevertheless, in this new version of the manuscript, we tried to put our results in the context of several models, particularly in Sect. 3.1 and also in Sect. 3.5, when discussing the peak altitude disagreement.

We note that we only mention CMAM in order to identify DE1 as the main contributor of k=2 but do not perform a direct comparison with CMAM.

- **Reviewer's comment:**
*5. What is the purpose of the GSWM/MSIS comparisons? What model version has been used and how? The given GSWM reference points to an old TIME-GCM study (where GSWM was used as a lower boundary condition only). There are several versions of GSWM around, the most recent one is GSWM-09 (see papers by Xiaoli Zhang). I doubt that this one has been used since no reference is given. Older GSWM versions had issues with seasonal variations and partly did not include the in-situ tidal forcing in the thermosphere. Also, GSWM is for 110 sfu (if I remember correctly) and does not include any solar flux dependence.*

**Author's response:**
The purpose is twofold: to see how well MIPAS ascending-descending zonal means (k=0 mode; DW1+TW3) compare with 10AM-10PM migrating tide fields in models and to evaluate how the migrating tides in the a priori are transferred to MIPAS temperatures  (the a priori acts as a vertically smoothing agent through the Tikhonov constraint in our retrievals). After this referee's comment, we understand that it is confusing to use two models for comparison. For consistency, we now only compare with one model at all altitudes. We chose MSIS, the a priori. The averaging kernels already pointed out that MIPAS thermospheric temperature measurement have a good quality (see Bermejo-Pantaleón et al., 2011), but the comparison presented here further supports that the retrieved temperatures do not contain significant information on model vertical structures. The comparison however shows a poor agreement between the measurements and the model. We have re-written the discussion accordingly.

We will postpone a thorough comparison with other models (not only GSWM) for a future work. Nevertheless, following another comment of this referee suggesting discussion in the context of models, we kept in the text the comparison with GSWM and also other models. We note that we now updated the GSWM results to GSWM-09 (Zhang et al., 2010a; Zhang et al., 2010b).

- **Reviewer's comment:**
*I am also puzzled to see that MSIS shows such a poor agreement with MIPAS. The MSIS amplitudes close to 150 km look way too small for migrating tides. Forbes et al. (2011, doi:10.1029/2011JA016855) compare the MSIS migrating diurnal tide at 400 km with CHAMP and GRACE. The agreement is actually quite good with amplitudes on the order of 120 K.*

**Author's response:**
We note that in the comparisons we have taken into account MIPAS sampling. MIPAS measurements provide the contribution of the DW1+TW3 only at 10AM. This coincides with the total migrating tide amplitude (which is the one shown in Forbes et al.) only at

the altitudes where the phase is 10AM. That is not the case of altitudes above 130km, where the in-situ diurnal tide, with a phase at 2-4PM in MSISE data, dominates. According to MSISE and as shown in the plot, DW1 contribution at 10AM is 15K at 150km (compared to 30K total MSISE DW1 amplitude).

We already mentioned this caveat in the text. We even wrote the factor of underestimation of the total DW1 amplitude. Nevertheless, we have re-written that paragraph for clarification.

- **Reviewer's comment:**
*6. Several conclusions are not supported by the data and speculation. (1) How do you know the propagation direction from the latitude/height Figure 7 (section 4.3, section 6)? Longitude/height plots give some indication about propagation direction, assuming that all tidal signals are propagating upward w/o any possible downward propagation or in-situ forcing (which is an assumption that needs to be stated!).*

**Author's response:**
We agree. We only know the tilt of the phase with altitude at certain latitude in the latitude/height maps of the phase (right hand side plots). The assumption of a vertical direction of propagation for proposing certain horizontal direction of propagation is now stated in the text in the introduction of Sect. 3 and also when used (several times along the manuscript). Following a suggestion of Referee#1, we now also include two figures where we plot the phase vs. altitude at selected latitudes (new Figs. 7 and 10). We hope this point is clearer in the manuscript now.

- **Reviewer's comment:**
*(2) The TW3 as the leading migrating component at 110 km (section 4.1, section 6) is mere speculation since MIPAS cannot separate DW1 and TW3. In-situ DW1 forcing is as likely (or more likely).*

**Author's response:**
We now make stress in section 4.1 and section 6 that DW1 is as likely.

- **Reviewer's comment:**
*7. Methodology section 3. I doubt that a non-expert in tidal satellite diagnostics will understand this section. It gives an overly complicated description of a well-established method that has been applied over the past 20 years to every single remote sensing infrared instrument when looking into tides. I strongly suggest to significantly shorten the section (or moving the shortened version into section 2 altogether). If the authors insist to keep this level of detail, the section should be moved into an appendix, but with the addition of a few intermediate steps that have been omitted, to help readers not familiar with the satellite orbit geometries and sampling.*

**Author's response:**
We moved old Section 3 to an Appendix. We have also re-written the section with the hope that it is more easily readable now.

Specific comments.
- **Reviewer's comment:**
*line 523. Oberheide et al. (2009) do not discuss the QBO in the westward propagating migrating tide, only in the eastward propagating DE3.*

**Author's response:**
We re-wrote the sentences and say now that Oberheide et al.'s referred to DE3.

- **Reviewer's comment:**
*The lower altitude in the Figures should be moved up to 50 or 70 km. There's not much tidal activity going on in the stratosphere.*
**Author's response:**
We moved lowest altitude of the Figures up to 40 km. We note that Zeng et al. (2008) detected DW1 activity (although with very small amplitudes 1K) already in the lower stratosphere.

- **Reviewer's comment:**
*The language is mostly fine but another round of proof-reading by the native speaker on the co-author list would be good.*
**Author's response:**
A native English speaker has proof-read the text.

**References**

Oberheide, J., M. G. Mlynczak, C. N. Mosso, B. M. Schroeder, B. Funke, and A. Maute (2013), Impact of tropospheric tides on the nitric oxide 5.3 um infrared cooling of the low-latitude thermosphere during solar minimum conditions, J. Geophys. Res. Space Physics, 118, 7283–7293, doi:10.1002/2013JA019278.

Oberheide, J., J. M. Forbes, X. Zhang, and S. L. Bruinsma Climatology of upward propagating diurnal and semidiurnal tides in the thermosphere *J. Geophys. Res., 116, A11306, doi:10.1029/2011JA016784, 2011.*

Zeng, Z., W. Randel, S. Sokolovskiy, C. Deser, Y.-H. Kuo, M. Hagan, J. Du, and W. Ward (2008), Detection of migrating diurnal tide in the tropical upper troposphere and lower stratosphere using the Challenging Minisatellite Payload radio occultation data, J. Geophys. Res., 113, D03102, doi:10.1029/2007JD008725.

---

## Author Response (AR2)

**Responses to comments to revised version**

We have taken into account the additional comments of the reviewer. Answers to his/her comments and a marked-up version of the manuscript are below.

**• Reviewer's comment**
*I am satisfied with the authors' responses and revisions to most of my comments. There are, however, one major comment and a few specific requests for clarification left that need to be addressed before the final version of the manuscript is published. All line numbers in the following refer to the annotated version of the manuscript (the one that was included in the author response pdf file).*

*Major comment:*

*line 548: It is puzzling that there is no QBO in the MIPAS wavenumber-4. This is inconsistent with SABER-based diagnostics of the DE3. The cited Li et al. paper does not quantify the QBO effect but uses vague language such as "slightly stronger" etc. Looking into Figure 4 of the Li et al. paper, I believe there's quite a bit of QBO like amplitude variability going on. Overall, Li et al., 2015, Figure 4 looks consistent with Oberheide et al., 2009, Wan et al., 2010, Zhou et al., 2015, who consistently found a 15-20% amplitude modulation by the QBO. This needs to be discussed more closely in the manuscript. At the very least, it needs to be stated that the MIPAS k=4 result (no QBO) is inconsistent with the the SABER results (which are internally consistent with TIDI tides) and QBO signals in the ionosphere (Wan paper, also recent Chang et al., COSMIC TEC diagnostics).*

**Author's response**
We state that we do not detect a QBO variablility in MIPAS DE3. However, we cannot say that there is no QBO at all. We actually derived some quasi-2-year variation in MIPAS DE3 (particularly around 110km; 2K QBO amplitudes) but within MIPAS errors and, additionally, correspondence with the stratospheric QBO phase is not clear at all altitudes. We deleted that 'the impact (...) is expected to be small' and the reference to Li et al., and include now some further discussion in the text.

Specific comments:

**• Reviewer's comment**
*In response to my original comment 2 to credit earlier work by Huang et al. re migrating tide-QBO studies, the authors now refer to Huang et al., 2006. I grant that some tidal discussion is included there, but the paper is more about the mean temperature. The more appropriate reference would be Huang et al., 2010 (10.1029/2009JD013698).*
**Author's response**
Reference changed.

**• Reviewer's comment**
*I am a little confused about the response to my original comment 3 (the solar flux dependence). The manuscript states (line 543) that for the inter-annual variability a solar flux component had to be included to get better results. Why is it then not "adequate" to perform an analysis for individual years from MIPAS?*
**Author's response**

We wanted to leave room for such a variation but we think that a thorough study would not be robust using only 5 years of data. Nevertheless, we have repeated the calculations without the solar component and the results barely change.

• **Reviewer's comment**
*Response to my original comment 5 (the part about the purpose of the GSWM/MSIS comparisons). line 269: GSWM-09 and CTMT do NOT include any in-situ tidal forcing in the thermosphere. Both models are designed to look into upward propagating tides and thus produce a (1,1) mode structure even at 110 km. However, at this altitude in-situ forcing kicks in. It needs to be clarified that both GSWM-09 and CTMT do not include the in-situ DW1. For GSWM-09, see for example the statement towards the end of section 1.3 in Zhang et al., 2010a. As such, the difference to MIPAS at 110 km is further support for the in situ forcing mentioned in the previous sentence in the manuscript.*
**Author's response**
We write now that GSWM-09 and CTMT do not include any in-situ tidal forcing.

• **Reviewer's comment**
*In the same context: line 283. I also believe that using tides from NRLMSISE below 110 km or so is not very meaningful, due to the data entering MSIS at these altitudes. Doug Drob and the NRL team work on a new MSIS version that will do a much better job. I urge the authors to omit the NRLMSIS data in the Figs between 80-100 km!!!! There is no use in comparing a model with real world data in a height region where the model is known to have deficiencies.*
**Author's response**
The main use of comparing MIPAS with MSIS is to check if tide in MSIS are being transfered to MIPAS retrievals. We include now a clarifying sentence.

• **Reviewer's comment**
*line 44: Oberheide et al. 2009 only discuss the DE3 QBO which most likely comes from the mesospheric QBO through the Ekanayake et al. mechanism.*
**Author's response**
We think that Oberheide et al. [paragraph 32]  also mention an alternative effect through stratospheric filetering during upward propagation, as suggested by Forbes and Vincent (1989).

• **Reviewer's comment**
*line 473: This is a very (too) strong statement. There is indeed a decent understanding of what's the DE3 is doing to the ionosphere, see for example recent papers by Loren Chang using COSMIC TEC. There are some unknowns, e.g., the SE2 contribution and possibly some uncertainty about how important F-region meridional winds are. I suggest to add a qualifying sentence.*
**Author's response**
We softened the statement.

• **Reviewer's comment**
*line 485: An additional contribution to the higher peak altitude could be a the interference from D0 and TE1. I suggest to mention this as an additional possibility.*
**Author's response**
We now write that interference with DW5 and/or TE1 is also possible.

After this comment, we noted that the contributing terdiurnal components were

misspelled for modes 1 and 4 (they were ok in Table 1). We corrected this.

• **Reviewer's comment**

*Figure 1 & 3, right column: figure titles seem to be printed over each other*

**Author's response**

The postcript figures look fine but, for some reason, they appear that way after compilation with latex. Let us please solve this problem during the manuscript production process if the problem persists.

• **Reviewer's comment**

*line 500: "There oscillation"*

**Author's response**

Changed.

[revised manuscript text omitted]